# A multimodal characterization of low-dimensional thalamocortical structural connectivity patterns
Alexandra John [1,2,3,4,5,6] ✉, Meike D. Hettwer [1,5,6,7], H. Lina Schaare [1,5,6], Amin Saberi [1,5,6], Şeyma Bayrak[1,5,6], Bin Wan [1,5,6], Jessica Royer [8], Boris C. Bernhardt [8] & Sofie L. Valk [1,5,6] ✉

The human thalamus is a heterogeneous subcortical structure coordinating whole-brain activity. Investigations of its internal organization reveal differentiable subnuclei, however, a consensus on subnuclei boundaries remains absent. Recent work suggests that thalamic organization additionally reflects continuous axes transcending nuclear boundaries. Here, we study how low-dimensional axes of thalamocortical structural connectivity relate to intrathalamic microstructural features, functional connectivity, and structural covariance. Using diffusion MRI, we compute a thalamocortical structural connectome and derive two main axes of thalamic organization. The principal axis, extending from medial to lateral, relates to intrathalamic myelin, and functional connectivity organization. The secondary axis corresponds to the core-matrix cell distribution. Lastly, exploring multimodal associations globally, we observe the principal axis consistently differentiating limbic, frontoparietal, and default mode network nodes from dorsal and ventral attention networks across modalities. However, the link with sensory modalities varies. In sum, we show the coherence between lower dimensional patterns of thalamocortical structural connectivity and various modalities, shedding light on multiscale thalamic organization.

As part of the diencephalon, the human thalamus is a bilateral gray matter structure that orchestrates whole-brain activity[1]. Due to its extensive connections to the entire cerebral cortex, and subcortical structures such as the basal ganglia and the cerebellum, the thalamus can be characterized as a 'connector hub' region in the brain. In the past, the thalamus was primarily described as a relay, passively transferring information to the cortex. More recent perspectives, however, are superseding this cortico-centric notion and emphasize the thalamic impact on mediating the corticocortical information transfer, the integration of information between cortical networks[2–4], and its role in cognition[5–9].

The internal organization of the thalamus is highly complex. Ex vivo, it has been parcellated into multiple subnuclei by analyzing histologically stained, dissected *post-mortem* brains[10,11]. These thalamic subnuclei contain a blend of two thalamic cell types, referred to as 'core' and 'matrix' cells, in varying proportions[12,13]. While core cells tend to target specifically layer IV

and V of primary sensory cortex, matrix cells have widespread cortical projections and innervate the supragranular layers I-III[12–15]. Further, the thalamic nuclei can be broadly classified into first-order nuclei (i.e., ventral posterolateral nucleus), that receive ascending sensory and modulatory cortical input, and higher-order nuclei (i.e., mediodorsal nucleus), that receive their input entirely from the cortex[16,17]. Compared to *post-mortem* studies, non-invasive neuroimaging using magnetic resonance imaging (MRI) provides the opportunity to acquire in vivo functional and structural data to study the thalamic organization and its relationship to the cortex in multiple modalities. While delineating the thalamus locally using standard T1- and T2-weighted (T1w, T2w) MRI images remains challenging due to poor tissue contrast[18], utilizing global approaches, such as structural and functional connectivity methods, that consider the extensive interrelation between the thalamus and the cerebral cortex has provided valuable insights into thalamic organization[19,20]. In the pioneering work of Behrens et al.[19], the

[1]Lise Meitner Research Group Neurobiosocial, Max Planck Institute for Human Cognitive and Brain Sciences, Leipzig, Germany. [2]International Max Planck Research School on Cognitive Neuroimaging (IMPRS CoNI), Max Planck Institute for Human Cognitive and Brain Sciences, Leipzig, Germany. [3]Brain Dynamics Graduate School, Leipzig University, Leipzig, Germany. [4]Faculty for Life Sciences, Leipzig University, Leipzig, Germany. [5]Institute of Neuroscience and Medicine (INM-7: Brain and Behaviour), Research Centre Jülich, Jülich, Germany. [6]Institute of Systems Neuroscience, Medical Faculty and University Hospital Düsseldorf, Heinrich Heine University Düsseldorf, Düsseldorf, Germany. [7]Max Planck School of Cognition, Leipzig, Germany. [8]Multimodal Imaging and Connectome Analysis Lab, Montreal Neurological Institute and Hospital, McGill University, Montreal, Canada. ✉e-mail: ajohn@cbs.mpg.de; valk@cbs.mpg.de

thalamus was parcellated based on thalamocortical (TC) probabilistic tractography using diffusion-weighted imaging (DWI)[19]. This parcellation was highly inter- and intra-subject reproducible[21] and showed robust correspondence with thalamic function[22]. Further approaches based on DWI were explored to uncover thalamic organization[23–25], and accordingly several thalamic atlases have been published[19,26,27]. Alongside structural connectivity, the functional coupling between the thalamus and cortex has been used to gain insights into the thalamic organization through TC functional connectivity[3,20,28,29]. Taken together, the existing literature suggests that the heterogeneous organization of the thalamus can be subdivided into nuclei dependent on scale and modality. These subnuclei vary in morphology, connectivity, and function.

Though subregions derived from different modalities overlap to a certain degree, a consensus of thalamic parcellation remains absent due to the high complexity of this region[30]. Specifically, drawing clear and universal boundaries proves challenging because of the complex connectivity profiles of thalamic nuclei, which target multiple cortical regions and are associated with various functional networks[3]. From a developmental point of view, the emergence of thalamic structural and functional properties is guided by molecular gradients of morphogens and transcription factors[31–33]. This is reflected in transitional patterns of gene expression and cytoarchiteture, transcending hard borders of thalamic nuclei[34–36]. Indeed, applying transcriptional profiling in mice revealed that gradual changes in gene expression are tied to anatomical and electrophysiological properties[36]. In line with this, recent advances in studying brain organization have shifted their focus on revealing spatially graded changes of neurobiological properties across the brain, in addition to the traditional approaches of defining discrete brain regions[37–42]. These continuous axes of spatial variation are referred to as 'gradients'. While this approach has mainly been applied to understand macroscale cortical organization, recent work uncovered transitional axes that help explain organizational patterns of the human thalamus[43–45]. These gradients, derived from TC structural and functional connectivity, might arise from smooth transition at the microscale level[34,36]. Based on the joint analysis of TC structural connectivity and gene expression data, a phylogenetically conserved medial-to-lateral axis has been reported, which captured transitions in cell type variations, and suggested a link to development and disease[43]. In addition, thalamic gradients derived from functional connectivity between the thalamus and cortex have been reported to follow a principal medial-to-lateral axis that was associated with thalamic gray matter volume, and a secondary anterior-to-posterior axis corresponding to functional networks[45]. Recently, functional TC gradients have been used to investigate the thalamic role in cortical organization in development[46]. From a cortico-centric perspective, it has been shown that the thalamic axis, reflecting variations in the spatial extent of corticothalamic structural connections, is linked to the sensory-association cortical hierarchy[47]. Taking a thalamo-centric perspective and drawing lines between internal organizational patterns of the thalamus and its interrelation to the cortex based on different modalities, may provide further valuable insights into this complex structure.

On top of genetic determination, thalamic functional activity has a substantial influence on cortical maturation through activity-dependent maturational processes[48,49]. A global measure used in neuroimaging that is hypothesized to capture both genetic and maturational coherences is structural covariance, which describes covariation in structural properties, such as cortical thickness, across different brain regions[50]. Although the biological mechanisms driving structural covariance remain incompletely understood, they may implicate activity-induced synaptogenesis and/or synchronous neurodevelopment[51,52], and relate to shared genetic effects[40,53]. In mice, TC structural covariance has been used to investigate thalamic organization, which showed some correspondence between TC structural and functional connectivity[54], suggesting a link between shared maturational patterning and connectivity in the thalamus.

The heterogeneous white matter connections between the thalamus and cortex, which form early in development and are biologically based on a heterogeneous distribution of structural entities, such as different thalamic cell types, are suitable as a direct measure to study thalamic organization[19,36,43,44,55]. In the current study, we explored how the internal organization of the human thalamus based on its structural connections to the cortex corresponds with the distribution of thalamic microstructural features, as well as TC functional connectivity and structural covariance. We assessed the structural connections between thalamic-seeds and the cortex by computing probabilistic tractography and extracted low-dimensional axes of thalamic organization. To contextualize our findings, we spatially associated these axes to intrathalamic microstructural features, such as the gray matter myelin based on quantitative T1 (qT1), and the distribution of cell types based on gene expression data. Additionally, we explored the association between the lower dimensional organization of structural and functional connectivity. Moving to a more global perspective, we further investigated how the thalamic gradients are related to macroscale cortical patterns and studied the link to TC functional connectivity and qT1-based TC structural covariance.

## Results

### Thalamic gradients based on TC structural connectivity (Fig. 1)
The first aim of this study was to investigate the spatial organization of the thalamus based on TC structural connectivity. To do so, we applied probabilistic tractography to map the white matter connections between thalamic voxels and 100 ipsilateral cortical parcels in each subject. The resultant thalamic-seed-by-cortical-parcel structural connectivity matrices contained the number of streamlines between seeds and parcels. After averaging across subjects ($N = 50$), the group-level structural connectivity matrix was normalized column-wise (Fig. 1A). To uncover the lower dimensions of this structural connectivity matrix, we first derived an affinity matrix (Fig. 1B), and further applied dimensionality reduction via diffusion map embedding. The decomposition yielded ten unitless gradient components, where nodes that share similar connectivity profiles were embedded closer together, and nodes with little connectivity similarity mapped further apart. Each gradient component represented a particular axis of the thalamic organization and explained a certain amount of variance (Fig. 1C). Henceforth, we focused on the first two components/gradients that accounted for a total variance of 46.42% in the left hemisphere and 46.47% in the right hemisphere. The principal TC structural connectivity gradient (G1$_{sc}$; explained variance left hemisphere (LH): 25.99%, right hemisphere (RH): 25.82%) defined a medial to lateral-central axis of the thalamus (Fig. 1D). The secondary gradient (G2$_{sc}$; explained variance LH: 20.43%, RH: 20.65%) located one apex at the medial-anterior but also posterior pole of the thalamus, and the opposite apex intersected the thalamus from anterior-lateral to medial-central (Fig. 1E). Further, we showed the robustness of gradient patterns derived from structural connectivity matrices, thresholded at different percentiles (Supplementary Fig. 1). Additionally, gradients 3 and 4 are presented in the supplementary materials (Supplementary Fig. 2).

To relate the TC structural connectivity gradients to previously defined discrete thalamic subnuclei, we used the THOMAS parcellation in MNI152 space[56,57]. In the 2D gradient frame, we color-coded data points according to the corresponding subnucleus, showing the spatial distribution of subnuclei along these two axes. Ordering the distinct nuclei based on the median of their gradient loadings revealed the following order for G1$_{sc}$: AV, MTT, Hb, MD, VA, Pul, MGN, VLa, VPL, VLP, CM (AV: Anterior ventral nucleus, MTT: Mammillothalamic tract, Hb: Habenular nucleus, MD: Mediodorsal nucleus, VA: Ventral anterior nucleus, Pul: Pulvinar nucleus, MGN: Medial geniculate nucleus, VLa: Ventral lateral anterior nucleus, VPL: Ventral posterior lateral nucleus, VLP: Ventral lateral posterior nucleus, CM: Centromedian nucleus[56]), and hence differentiated between higher-order nuclei and sensorimotor-projecting nuclei. Ordering the nuclei along G2$_{sc}$ resulted in: MGN, VPL, AV, Pul, CM, VLP, Hb, MD, MTT, VA, VLa, and henceforth does not separate first-order and higher-order nuclei (Fig. 1F). Here, results of the left hemisphere are reported, however, they were similarly replicated in the right hemisphere (Supplementary Fig. 3).

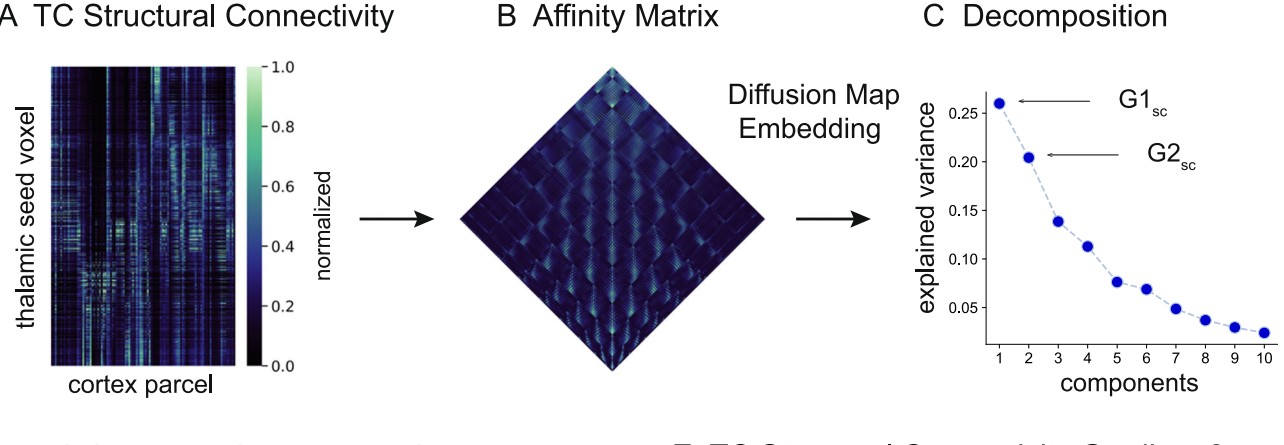

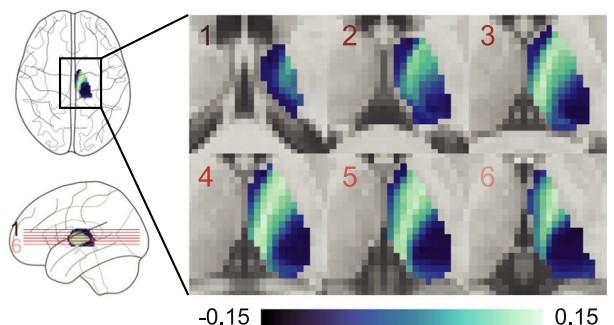

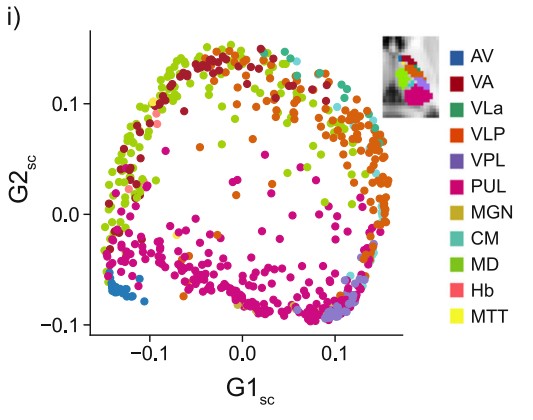

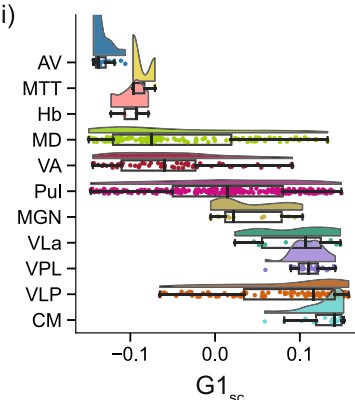

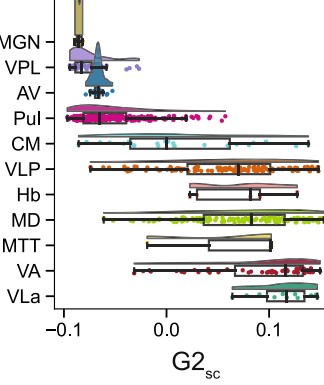

**Fig. 1 | Thalamocortical structural connectivity gradients. A** Normalized group-level structural connectivity matrix ($N = 50$), resulting from probabilistic tractography computation between thalamic seed-voxels and cortical parcels (i.e., TC). **B** Affinity matrix derived from group-level structural connectivity matrix using a normalized angle similarity kernel. **C** Decomposition of affinity matrix into ten gradient components using diffusion map embedding. For each component, the corresponding explained variance is displayed. **D** Gradient loadings of component 1 ($G1_{sc}$) projected onto the thalamus (axial planes). The red lines in the glass brain indicate the position of each respective axial slice of the displayed thalamus. **E** Gradient loadings of component 2 ($G2_{sc}$) projected onto the thalamus. Slice positions are congruent to (**D**). **F** Decoding of $G1_{sc}$ and $G2_{sc}$ based on THOMAS atlas. (i) 2D space framed by $G1_{sc}$ and $G2_{sc}$. Each data point represents a thalamic voxel, color-coded by the thalamic subnucleus to which it belongs. (ii) Raincloud plots display the gradient loadings of $G1_{sc}$ and $G2_{sc}$ per nucleus and are ordered by median, respectively. The boxes represent the interquartile range (25th to 75th percentile), lines depict medians, and whiskers are defined by values 1.5 times the interquartile range. All results are presented for the left hemisphere; however, they were similarly replicated in the right thalamus (Supplementary Fig. 3). AV anterior ventral nucleus, VA ventral anterior nucleus, VLa ventral lateral anterior nucleus, VLP ventral lateral posterior nucleus, VPL ventral posterior lateral nucleus, Pul pulvinar nucleus, MGN medial geniculate nucleus, CM centromedian nucleus, MD mediodorsal nucleus, Hb Habenular nucleus, MTT Mammillothalamic tract.

## TC structural connectivity gradients are associated with microstructure and functional connectivity (Fig. 2)

After computing structural connectivity gradients, we probed whether they relate to underlying thalamic organizational properties, such as microstructural features (distribution of qT1 values as a proxy for myelin, distribution of core and matrix cells), and thalamic gradients based on TC functional connectivity. Except for the core-matrix map, all measures were derived from the same dataset ($N = 50$).

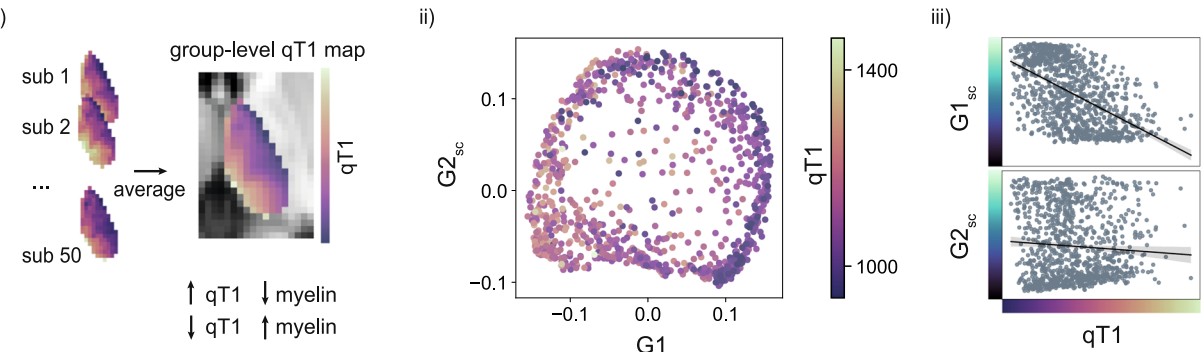

**A** Association of Structural Connectivity Gradients with Group-Level Thalamic qT1

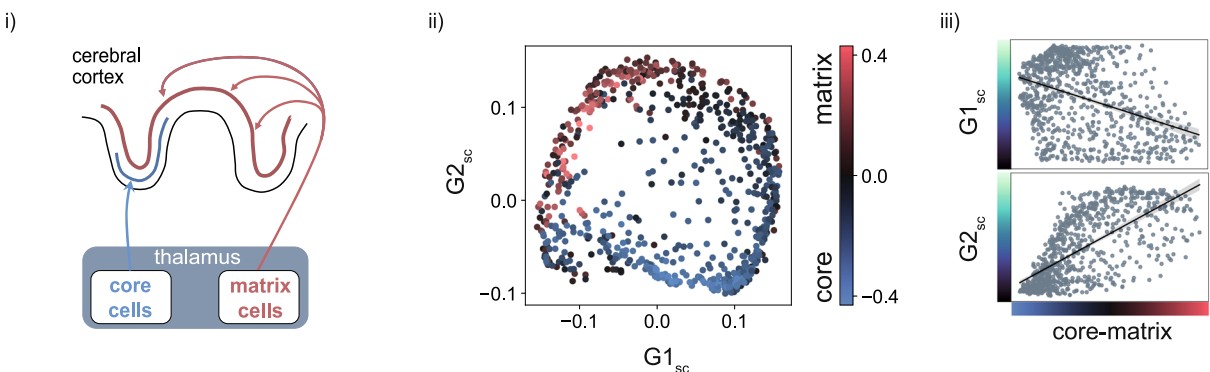

**B** Association of Structural Connectivity Gradients with Core-Matrix Cell Distribution

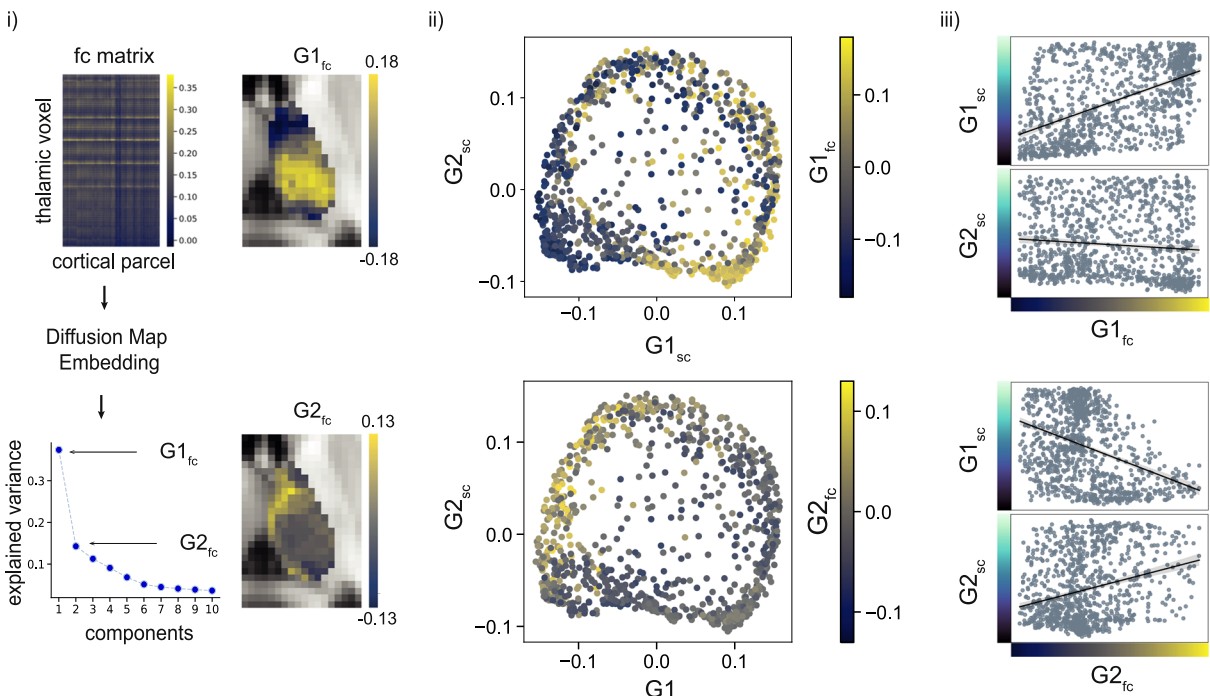

**C** Association of Structural Connectivity Gradients with Functional Connectivity Gradients

**Intrathalamic myelin.** To examine whether $G1_{sc}$ and $G2_{sc}$ reflect intra-thalamic microstructural variation, we used the group-level thalamic qT1 values as a modality estimate for gray matter myelin (Fig. 2A). Note that lower qT1 values are associated with a higher myelin content and vice versa. We tested the statistical relationship between both gradient maps and the group-level qT1 map using Pearson correlation, and corrected for spatial autocorrelation using variograms[58]. This analysis suggested a link between $G1_{sc}$ and the spatial distribution of qT1 (LH: r = −0.536, $p_{SA}$ = 0.038), whereas there was no significant correlation between $G2_{sc}$ and the qT1 map (LH: r = −0.068, $p_{SA}$ = 0.873). This trend was replicated in the right thalamus (RH: $G1_{sc}$: r = −0.594, $p_{SA}$ = 0.011; $G2_{sc}$: r = 0.119, $p_{SA}$ = 0.794).

**Fig. 2 | Contextualization of gradients with microstructure and functional connectivity.** **A** Contextualization with quantitative T1 (qT1). (i) Individual thalamic qT1 values were averaged to create a group-level qT1 map ($N = 50$). Note the inverse relationship between qT1 intensity and approximated gray matter myelin (represented by black arrows). (ii) 2D space framed by structural connectivity gradient 1 ($G1_{sc}$) and 2 ($G2_{sc}$). Data points represent thalamic voxels, color-coded by thalamic group-level qT1 intensity. (iii) Correlation between qT1 intensity and $G1_{sc}$, as well as qT1 intensity and $G2_{sc}$. **B** Contextualization with core-matrix distribution. (i) Conceptualized representation of the core-matrix framework. Core cells (blue) project in a specific fashion to granular layers of the cerebral cortex, whereas matrix cells (red) innervate superficial cortex layers in a distributed fashion. (ii) 2D space framed by $G1_{sc}$ and $G2_{sc}$. Data points represent thalamic voxels, color-coded by the core-matrix difference map. Negative values of the colormap (blue) indicate a higher proportion of core cells, whereas positive values (red) indicate a higher proportion of matrix cells. (iii) Correlation between core-matrix difference map and $G1_{sc}$, as well as core-matrix difference map and $G2_{sc}$. **C** Contextualization with functional connectivity gradients. (i) Group-level ($N = 50$) functional connectivity (fc) matrix (z-scored), resulting from correlating thalamic voxel and cortical parcel time-series, followed by gradient decomposition into 10 components with respective eigenvalues. Principal and secondary functional connectivity gradients ($G1_{fc}$ and $G2_{fc}$) displayed on the axial thalamus slice (see Supplementary Fig. 4 for the right hemisphere). (ii) 2D space framed by $G1_{sc}$ and $G2_{sc}$. Data points represent thalamic voxels, color-coded by $G1_{fc}$ loadings (top) and $G2_{fc}$ loadings (bottom). (iii) Correlation between $G1_{fc}$ and structural connectivity gradients $G1_{sc}$ and $G2_{sc}$ (top), and $G2_{fc}$ and structural connectivity gradients $G1_{sc}$ and $G2_{sc}$ (bottom); All results are displayed for the left hemisphere.

**Distribution of core and matrix cells.** Another main feature of thalamic organization on a microscale level is the varying distribution of core cells and matrix cells, which also differ in their TC projection patterns[10,12]. To probe whether our identified gradients mirror this distribution, we used a difference map capturing the proportion of core cells and matrix cells based on messenger ribonucleic acid (mRNA) level estimates[59]. Statistically testing the relationship between the core-matrix map and $G1_{sc}$ (LH: $r = -0.378$, $p_{SA} = 0.135$) and $G2_{sc}$ (LH: $r = 0.676$, $p_{SA} = 0.044$) in the left hemisphere suggested an association between the distribution of core- and matrix cells and $G2_{sc}$ (Fig. 2B). This analysis was limited to the left hemisphere due to the small sample size on which the right core-matrix map was grounded (see Methods).

**TC functional connectivity gradients.** Next, we explored the link between the low-dimensional organization of structural and functional TC connectivity (Fig. 2C). Therefore, we calculated the TC functional connectivity by correlating the resting state time series of thalamic seed voxels and cortical parcels and averaged across subjects. Analog to the computation of structural connectivity gradients, we applied diffusion map embedding to uncover the lower dimensional organization of the group-level TC functional connectivity. We correlated the resulting principal and secondary functional connectivity gradient maps ($G1_{fc}$ and $G2_{fc}$, RH shown in Supplementary Fig. 4) with the structural connectivity gradients, and corrected for spatial autocorrelation using variograms. In both hemispheres, $G1_{fc}$ was correlated with $G1_{sc}$ (LH: $r = 0.526$, $p_{SA} = 0.044$, RH: $r = 0.564$, $p_{SA} = 0.014$). $G1_{fc}$ was not related to $G2_{sc}$ (LH: $r = -0.095$, $p_{SA} = 0.872$, RH: $r = -0.177$, $p_{SA} = 0.754$). The analysis further did not show an association of $G2_{fc}$ with $G1_{sc}$ (LH: $r = -0.374$, $p_{SA} = 0.083$, RH: $r = -0.133$, $p_{SA} = 0.532$) and $G2_{sc}$ in the left hemisphere ($r = 0.265$, $p_{SA} = 0.242$) but was correlated with $G2_{sc}$ in the right hemisphere ($r = 0.484$, $p_{SA} = 0.016$).

Taken together, we could show that TC structural connectivity gradients differentially reflected microstructure and cellular properties. Further, we observed an association between TC structural connectivity gradients and intrinsic TC functional connectivity organization. To evaluate consistency, we present the Pearson correlation (r) values between individual structural connectivity, qT1, and functional connectivity maps and their respective group-level maps. The findings indicate high consistency for the principal structural connectivity gradient and qT1, while functional connectivity gradients show the most variation across individuals (Supplementary Fig. 5). Additionally, to ensure thoroughness, we examined whether structural gradients 3–10 were related to the group-level qT1 map, the core-matrix distribution, or TC functional gradients 1 and 2. Overall, we observed mostly spurious effects, except for a bilateral association between $G4_{sc}$ and $G2_{fc}$ (LH: $r = 0.388$, $p_{SA} = 0.013$; RH: $r = 0.367$, $p_{SA} = 0.018$), as well as few unilateral associations such as between $G10_{sc}$ and the core-matrix map (LH: $r = 0.324$, $p_{SA} = 0.020$) (Supplementary Table 1). Additionally, in a supplementary analysis, we explored the association between functional connectivity gradients and the group-level qT1 and core-matrix maps, revealing a correlation between $G2_{fc}$ and the core-matrix map (LH: $r = 0.568$, $p_{SA} = 0.01$) (Supplementary Table 2). Finally, to test the

robustness of our results, we compared the TC structural connectivity gradients reported here to previous work on joint thalamic gene expression and structural connectivity[43]. Overall, we found that both $G1_{sc}$ and $G2_{sc}$ were spatially associated with the main gradient reported in Oldham and Ball[43] (LH: $G1_{sc}$: $r = 0.478$, $p_{SA} = 0.037$; $G2_{sc}$: $r = -0.701$, $p_{SA} = 0.039$). When examining the relation between $G1_{sc}$ and $G2_{sc}$ and the gradients based on structural connectivity and gene expression separately, we observed a clear differentiation, with our $G1_{sc}$ tending to relate to gene expression (LH: $r = 0.423$, $p_{SA} = 0.054$) but not structural connectivity (LH: $r = -0.061$, $p_{SA} = 0.778$), while $G2_{sc}$ was associated to structural connectivity based gradient of Oldham and Ball[43] (LH: $r = -0.817$, $p_{SA} < 0.001$) but not transcriptomic patterns (LH: $r = -0.585$, $p_{SA} = 0.227$) (Supplementary Table 3).

## Cortical projections of structural connectivity gradients and their associations to functional connectivity and structural covariance (Fig. 3)

Next, we explored the TC associations based on white matter connectivity, functional connectivity, and structural covariance to understand how thalamic and cortical organization interrelate.

**Cortical projection of TC structural connectivity gradients.** First, we computed for each cortical parcel the correlation between the TC structural connectivity profile and $G1_{sc}$. The resulting Pearson's r value was projected onto the equivalent cortical parcel (Fig. 3A). Negatively loaded cortical regions indicated a stronger relationship with the thalamic medial portion, whereas positively loaded regions indicated a stronger relation with the lateral-central subregions of the thalamus. To decode the macroscale cortical patterns, the correlation coefficients were grouped based on functional communities[60] and sorted according to the mean r values per community. The decoding revealed a trend following the functional hierarchy from limbic to somatomotor networks with the visual network deviating from this trend. It is worth noting that the lateral geniculate nucleus (LGN, visual projections) is not included in the thalamic mask due to its peripheral localization. Mapping of $G2_{sc}$ onto the cortical surface followed the same principle and resulted in a dissociation between posterior and anterior cortical areas. More specifically, negative loadings were located in the somatomotor, and dorsal attention network, whereas positive loadings were situated at the prefrontal cortex and cingulum (Fig. 3A). Patterns were reproduced for the right hemisphere (Supplementary Fig. 6). Summed up, the principal structural connectivity gradient mapped onto the cortex differentiated somatomotor and limbic areas, while the secondary gradient differentiated between anterior and posterior cortical regions.

**Associations of TC structural connectivity gradients with TC functional connectivity.** Next, to study the association between structural connectivity gradients and functional connectivity, we correlated $G1_{sc}$ and $G2_{sc}$ with the functional connectivity profiles of each parcel. The resulting Pearson correlation coefficients were projected onto the cortical surface (Fig. 3B). For $G1_{sc}$, negative loadings that indicate a

## A  Structural Connectivity Gradients Mapped onto Cortex

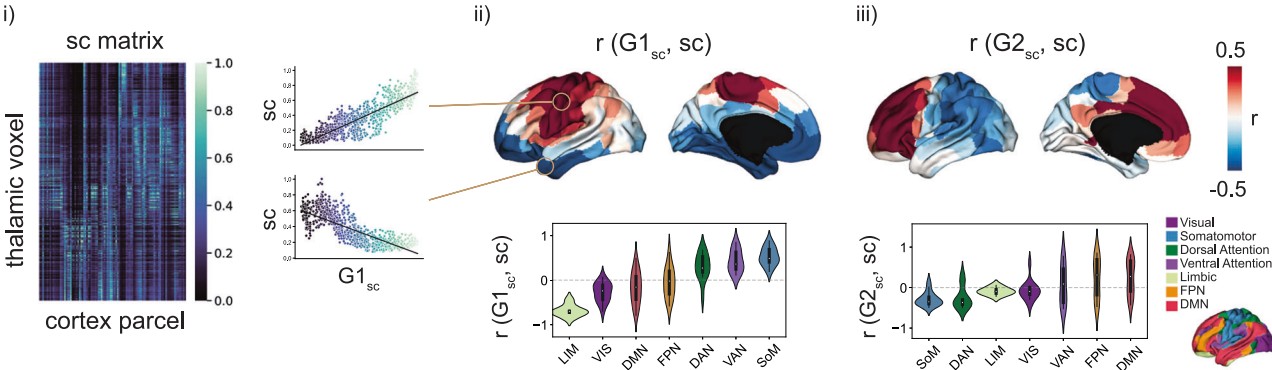

## B  Parcel-wise Correlation of Structural Connectivity Gradients and Functional Connectivity

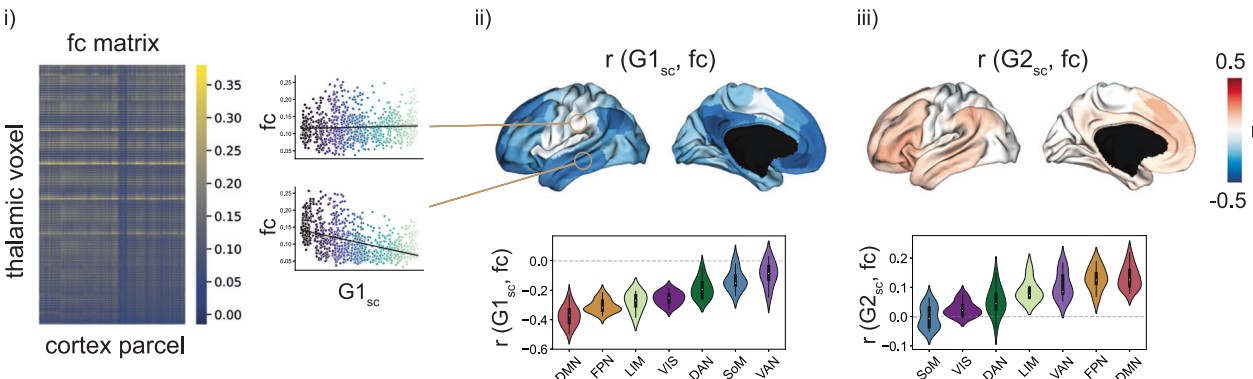

## C  Parcel-wise Correlation of Structural Connectivity Gradients and Structural Covariance

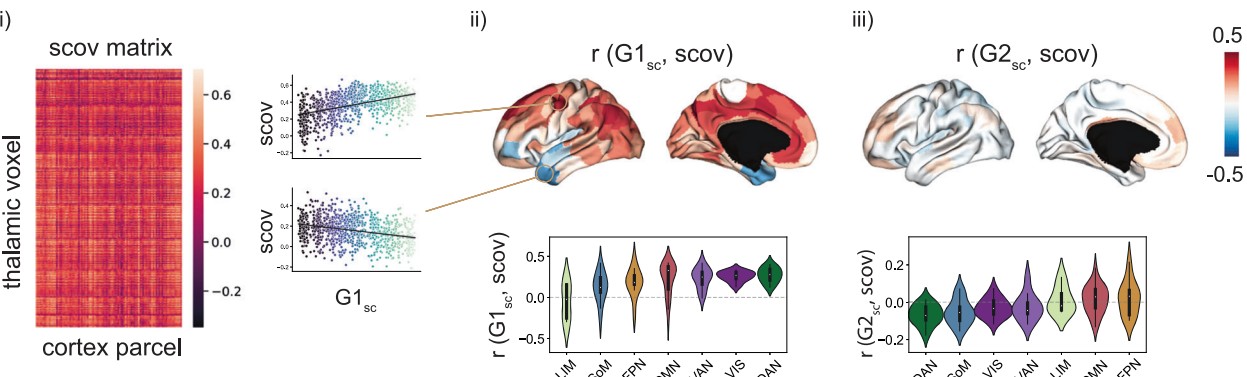

**Fig. 3 | Cortical projections of structural connectivity gradients and their associations to functional connectivity and structural covariance. A** Structural Connectivity (sc). (i) Sc matrix and Pearson correlation of sc profiles of each parcel (schematically displayed for 2 parcels) with the principal structural connectivity gradient (G1$_{sc}$). (ii) Projection of r values resulting from parcel-wise correlation between sc profiles and G1$_{sc}$ onto the cortex. Thus, negative values (blue) indicate relation with the medial part of the thalamus, whereas positive values (red) indicate relation with lateral thalamic portions. Decoding of cortical pattern leveraging functional communities (ordered along mean). (iii) Analog parcel-wise correlation between sc profiles and secondary structural connectivity gradient (G2$_{sc}$).
**B** Functional Connectivity (fc). (i) Fc matrix and Pearson correlation of resting state fc strength of each parcel (schematically displayed for 2 parcels) with G1$_{sc}$. (ii) Projection of r values resulting from parcel-wise correlation between fc profiles and G1$_{sc}$ onto the cortex, and decoding of cortical pattern leveraging functional communities (ordered along mean). (iii) Analog parcel-wise correlation between fc profiles and G2$_{sc}$. **C** Structural Covariance (scov). (i) Scov matrix and Pearson correlation of thalamocortical scov of each parcel (schematically displayed for 2 parcels) with G1$_{sc}$. (ii) Projection of r values resulting from parcel-wise correlation between scov profiles and G1$_{sc}$ onto the cortex, and decoding of cortical pattern leveraging functional communities (ordered along mean). (iii) Analog parcel-wise correlation between scov profiles and G2$_{sc}$. Results shown are at group-level (N = 50). Violin plots represent kernel density estimates of the data. The boxes indicate the interquartile range (25th to 75th percentile), the white dots represent the median, and the whiskers extend to 1.5 times the interquartile range. FPN Fronto-parietal Network, DMN Default Mode Network.

strong relation with the medially located apex of G1$_{sc}$ were present in regions associated with the default mode network (DMN) but also frontoparietal network (FPN), and limbic networks. Positive correlations between parcel-wise functional connectivity profiles and G1$_{sc}$,

that would indicate a relation to the more lateral-central thalamic apex, were almost not present (LH: 3% of parcels in range $0 \leq r \leq 0.03$, RH: 8% of parcels in range $0 \leq r \leq 0.09$). For G2$_{sc}$ and parcel-wise functional connectivity profiles, negative correlation

coefficients were generally low (LH: 10% of parcels in range $-0.05 \leq r \leq 0$, RH: 1% of parcels in range $r \leq 0$). Positive correlation coefficients were present in regions of DMN, FNP, and ventral attention network (VAN). Patterns were reproduced for the right hemisphere (Supplementary Fig. 6).

**Associations of TC structural connectivity gradients with TC structural covariance.** Last, we aimed to descriptively assess to what extent the defined structural connectivity gradients reflect patterns of TC structural covariance, a global measure that is hypothesized to capture maturational coherence[51]. Therefore, we computed a TC structural covariance matrix by correlating the qT1 values of each thalamic voxel with the qT1 value of each cortical parcel (collapsed along cortical depth) across subjects ($N = 50$). Next, we correlated G1$_{sc}$ and G2$_{sc}$ with the parcel-wise structural covariance profiles and mapped the results onto the cortical surface (Fig. 3C). For G1$_{sc}$, negative correlations indicate stronger covariance between medially located thalamic areas and the temporal pole, lateral frontopolar cortex, and parts of the superior temporal gyrus. Positive loadings showed a dispersed pattern in superior parts of the cerebral cortex, where the highest correlations were present in superior frontal, parieto-occipital regions, precuneus, and cingulate gyrus associated with lateral portions of the thalamus. Highest loadings were found in the dorsal attention network (DAN). Here, results are reported for the left hemisphere (Fig. 3C). The pattern in the right hemisphere (Supplementary Fig. 6) deviated partially (i.e., positive loadings in the superior temporal gyrus), possibly linked to asymmetries in cortical microstructural variability across individuals (Supplementary Fig. 7). For G2$_{sc}$ correlation coefficients were generally low (LH: $-0.18 \leq r \leq 0.21$, Fig. 3C; RH: $-0.28 \leq r \leq 0.22$; Supplementary Fig. 6).

Last, to assess consistency of gradient observations, we selected three prominent example nuclei of the THOMAS atlas (AV, VPL, and MD) that are distributed along G1$_{sc}$, and explored the structural and functional connectivity projections, and structural covariance patterns. Again, we found the clearest differentiation of thalamic cortical projections for structural connectivity, followed by functional connections. Covariance showed most deviating patterns from the known projection patterns of the nuclei, as well as interhemispheric differences (Supplementary Fig. 8).

## Discussion

In this study, we aimed to characterize the thalamic organization based on its structural connectivity profiles to the neocortex, beyond the level of distinct nuclei borders. We followed a multimodal approach to explore the association between the defined axes of structural connectivity based thalamic organization and intrathalamic microstructural properties, TC functional connectivity, and TC structural covariance, a proxy for maturational coherence. Our study reported a principal axis/gradient of thalamic organization spanning from medial-to-lateral portions and segregating thalamic higher-order nuclei and sensorimotor-projecting nuclei. This pattern spatially correlated with the intrathalamic qT1 distribution, which we used as a proxy for microstructure, and with the first gradient based on TC functional connectivity. Mapped onto the cortex, the principal structural connectivity gradient separated limbic regions and somatomotor regions. Evaluating the multimodal convergence of TC associations (structural connectivity, functional connectivity, and structural covariance), we observed for the principal axis a differentiation of cortical limbic, FPN, and DMN regions from DAN and VAN across modalities. Notably, associations with visual and sensorimotor regions were inconsistent and might arise from modality-specific differences or methodological noise. The secondary axis/gradient reflected the spatial distribution of core and matrix cells and was associated with the secondary gradient based on TC functional connectivity in the right hemisphere. Mapped on the cortex, the secondary gradient differentiated cortical anterior from posterior regions, showed associations between the positive gradient apex and functional connectivity strength in the FPN and DMN, while generally displaying a relatively weak association with structural covariance.

We acknowledge the diverse methods used to study the thalamus, from immunochemistry and histology to MRI-based parcellations, each contributing valuable insights while facing unique challenges, such as obtaining sufficient high-quality specimens for detailed histological analysis, and varying modality-specific MRI parcellation schemes. Our research uses a gradient decomposition approach, which reveals relationships between thalamic areas independent of their spatial location, highlighting similarities in structural connectivity patterns, and its coherence with microstructure, and functional connectivity. Importantly, we also expand the focus to examine the thalamus in relation to the cortex, offering a broader perspective on brain organization. Our multimodal approach synthesizes different methodologies, advancing the understanding of thalamic organization in a more integrated and comprehensive way.

Overall, we identified two axes of organization that align with microstructural patterns (myelin, core-matrix), functional connectivity, and, to some extent, structural covariance, which serves as a proxy for shared genetic patterning and maturation[51]. In particular, our findings revealed a principal medial-to-lateral thalamic axis/gradient based on TC white matter connectivity. To link the gradients to their underlying thalamic nuclei composition, we used a well-established atlas for decoding[57] and observed that most nuclei tend to localize at specific positions along either both gradients (e.g., VPL, AV) or at least one (e.g., Pulvinar along G2$_{sc}$). Although some nuclei, such as the Pulvinar, showed dispersion along the principal axis, limiting clear pattern identification along this axis, a trend emerged in G1$_{sc}$ where nuclei projecting to association cortices (higher-order nuclei such as AV, MD, VA) cluster at the medial pole of the gradient, and those projecting to sensorimotor cortex (e.g., VPL, VLa, MGN, CM) appeared at the lateral pole. Note, the CM, though a small intralaminar nucleus, is classified as a higher-order nucleus but also projects to motor regions[61,62]. We derived that the main variations in TC connectivity profiles map an axis differentiating sensorimotor-projecting regions and transmodal regions, which is also consistent with the pattern observed when the principal axis was projected onto the cortex, revealing a clear division between limbic and somatomotor cortical regions. Our observations broadly align with recent work using a joint analysis of TC structural connectivity and gene expression data[43], which reported a similar medial-to-lateral axis, with the difference that in our case the medial apex tended to be more pronounced toward the center of the thalamus. The hierarchical representation of projection patterns was also found in the mouse thalamus after computing an organizational axis based on tract tracing and gene expression data, spanning from somatosensory regions to lateral/frontal regions, and may suggest the hypothesis that this axis is phylogenetically-conserved[43]. Further, the medial-to-lateral axis may originate from the spatiotemporal development of the brain. It has been shown that the formation of connectivity-based subdivisions of the human thalamus expanded from lateral to medial portions during the perinatal period[44]. Moreover, a study in mice has shown a thalamic medial-to-lateral pattern, with a topological shift across nuclei borders in gene expression and electrophysiological properties[36]. Next to a medial-to-lateral axis, we observed a second axis of organization with one apex located at the medial-anterior and posterior pole of the thalamus, and the opposite apex intersecting the thalamus from anterior-lateral to central-medial. Decoding of the secondary gradient with the atlas did not yield a clear pattern and segregated MGN, VPL, AV, Pulvinar, CM, and VLP from Hb, MD, MTT, VA, VLa regions on top of the differentiation observed in the first gradient. Reviewing animal studies[34], we suggest that the heterogeneity of thalamic organization exceeds distinct nuclei borders and can be described along gradual axes. In sum, our work revealed patterns of thalamic organization along more than one axis, with the principal axis capturing a distinction of transmodal and unimodal differences in projection patterns.

Having established two axes of intrathalamic structural organization using structural connectivity that differentiate thalamic subareas, we further explored how these axes were associated with microstructure, as probed by qT1, a proxy for gray matter myelin. Indeed, we could show that the principal medial-to-lateral axis corresponded to variations in the intrathalamic myelin profile, where lateral and hence thalamic sensorimotor regions

showed higher myelination compared to thalamic higher-order regions. This observation aligns with a recent finding suggesting a higher proportion of oligodendrocytes in the lateral portion of the thalamus[43]. The distribution of myelin could again touch on the spatiotemporal brain development during the perinatal phase. Zheng et al. [44] reported a lateral-to-medial development of thalamic microstructure, where fiber integrity (measured by fractional anisotropy, fiber density, and diffusivity) in the lateral thalamus seemed to develop faster compared to medial thalamic portion. Further, our finding of higher myelination in lateral thalamic portions, which captured unimodal nuclei, resonated with observations of higher myelination of unimodal regions in the cerebral cortex[63] and may be related to a higher conduction velocity in sensorimotor regions compared to transmodal regions[64,65]. Second, in order to further investigate the relationship to thalamic microstructure, we leveraged a map that indicates the weighting of core versus matrix cells in thalamic voxels, that was created based on mRNA expression levels of the calcium-binding proteins Parvalbumin and Calbindin[59]. We found the secondary axis, but not the principal axis, was correlated with the cell type distribution. This axis differentiated cortical anterior from posterior regions. Notably, the second functional gradient was also correlated with the cell type distribution, suggesting that the core and matrix cells form the basis for both structural and functional organizational patterns. Of note, due to the imbalanced donor distribution (6 donors of the left versus 2 of the right hemisphere), we only tested this association in the left hemisphere. Further work will be needed to more precisely map the distribution of core and matrix cells in the human thalamus using a larger sample size. We further observed a relation between the low dimensional organization of the principal structural and the principal functional TC connectivity gradient, and in one hemisphere, the secondary structural and secondary functional gradient. The functional connectivity pattern in part recapitulated observations of Yang et al. [45] and together may point to multiple differential axes of organization within the thalamus. While demonstrating an overlap of organizational principles across modalities, differences between patterns of structural and functional connectivity are expected due to method-specificities (i.e., functional connectivity arising not only from direct but also indirect connections). Comparing our work to previously published patterns from Oldham and Ball[43], we found that both our gradients related to the joint structural connectivity and transcriptomic TC gradient and differentiate in patterns based on structural connectivity ($G2_{sc}$) and transcriptomic expression ($G1_{sc}$) separately. Though somewhat surprising, we believe these differences strengthen the validity of our observed patterns. Both gradients show spatial alignment with the main organizational features identified by an independent research group, while also demonstrating distinct associations - one with genetic expression and the other with structural connectivity. Our argument that the thalamus may contain multiple meaningful spatial patterns therefore holds and may extend previous work focusing on a single gradient across modalities. The difference between both patterns may relate to the previously discussed differentiation between microstructural patterns (linked to $G1_{sc}$) - and thus in addition related to gene expression, whereas core-matrix patterns (linked to $G2_{sc}$) - relate primarily to structural connectome patterns in the Human Connectome Project sample. Each modality in our study has a distinct methodological approach that may hint at a differentiable biological origin: structural connectivity reflects white matter tracts; functional connectivity relates to correlations between bold time-series, thought to reflect direct and indirect connectivity; and qT1 reflects local patterning of microstructure, linked to myelination[66]. While these features are ultimately interrelated to some extent, we propose that they may reflect differentiation into distinct biological pathways, which could align with the varied definitions of nuclei boundaries across different modalities.

Indeed, also in parcellation approaches, it has been shown that DTI based clusters correspond higher with structural parcellations in larger nuclei, while parcellations based on resting-state functional MRI (rsfMRI) agree more with structural parcellation in smaller nuclei[30]. Generally, the observation that different organizational dimensions within the thalamus correspond to different structural and functional features indicates that the

thalamus and its subnuclei can be differentiated based on different neurobiological principles. It is possible that the divergence observed between the principal and secondary gradient relates to differentiable influences of activity-dependent plasticity and maturation (based on the association of the principal gradient with myelin-proxy profiles), whereas the secondary gradient reflects a different, yet related, organization of core and matrix cells that is scaffolding development but not malleable. However, further work is needed to understand and test the association between cell-level differentiation and possible maturation-related myelination profiles in the thalamus. Our study accessed human thalamic organization at the macroscale, revealing two gradients and combining several modalities. Numerous studies have also demonstrated cortical gradients across genetic, maturational, structural, and functional dimensions[37,40–42,67], suggesting that similar gradient patterns in the thalamus may reflect cortical connectivity, though causation remains unclear.

Lastly, we aimed to explore how defined patterns of thalamic organization are interrelated with cortical patterns based on structural connectivity, functional connectivity, and structural covariance. Evaluating the cortical projections of the principal structural connectivity gradient revealed a differentiation between limbic functional networks, which were mostly associated with medial thalamic portions, and the somatomotor network, which was related to the lateral thalamic portions. A comparable pattern was observed in functional connectivity. The overall distinction of cortical somatomotor and transmodal projections is in line with the pattern that was revealed by ordering the thalamic nuclei along the principal gradient, demonstrating a trend from higher-order nuclei to sensorimotor-projecting nuclei. Of note, in the cortical projection patterns, we found the visual network closer at the gradient apex that was in our model assigned to transmodal regions and was therefore deviating from the sensorimotor-association functional hierarchy. This observation may arise due to the exclusion of the LGN from our thalamic mask (projects to primary visual cortex[10,68]), and further may be driven by the projections of the Pulvinar encompassing higher-order areas and the visual system[69]. The distinction between sensorimotor and association areas based on structural and functional connectivity aligned with previous work in humans and animals[13,47,70,71] that illustrated how the thalamus may be a key node in the brain, coordinating both sensorimotor and abstract cognitive functions[1,72]. Moreover, the cortical projection pattern of the principle structural connectivity gradient echoed maps of laminar differentiation and sensory-transmodal axes, patterns possibly linked to cortical maturation during the first two decades of human development[38,65,73,74]. Also, this cortical pattern aligned to some extent with notions describing a sensory-fugal gradient of cytoarchitectural complexity[41,75], where sensory areas have koniocortical characteristics of a well-developed granular layer IV and fugal or paralimbic areas exhibit a dys/agranular cytoarchitecture. Aggregating our findings and the mentioned literature, the principal gradient derived from TC structural connectivity may in both, the thalamus and the cortex, be related to development, microstructure, and functional hierarchies.

The cortical projection of the secondary structural connectivity gradient revealed a combination of cortical posterior regions (including somatomotor, DAN, limbic, and visual regions) with the negative thalamic apex (medial-anterior and posterior pole of the thalamus), and cortical anterior regions related to the positive thalamic apex (intersection from anterior-lateral to central-medial). Though less pronounced, the pattern was to some extent reflected in TC functional connectivity. The cortical projection of the secondary axis substantially resembled the cortical pattern reported in Müller et al. [59] that represents the correlation between TC functional connectivity strength and the relative distribution of core and matrix cells, which was consistent with our observation that the secondary structural connectivity gradient is correlated with the relative difference of core and matrix cells in the thalamus. Thalamic regions that contain relatively higher proportions of core cells showed preferential functional coupling to somatosensory cortices, while regions with higher matrix cell proportions preferentially couple with transmodal cortical regions[59]. Additionally, they found that matrix cell regions tend to couple to cortical

areas with a lower intrinsic timescale[59]. Thus, overall both gradients pointed to a differentiation between sensorimotor and transmodal networks, mirroring observations in recent work of a sensorimotor and association 'motif' of thalamic cortical patterning. Extending this work, we illustrated how these motifs may be embedded within the intrinsic organization of the thalamus along two main structurally defined gradients. Moreover, we found high correspondence between cortical projections of the structural connectivity gradients and functional connectivity patterns.

Throughout the course of development, the thalamus and cortex are closely interconnected. Therefore, we probed whether regions that share similarities in TC connectivity are associated with structural covariance, which has been suggested to reflect shared maturation[51]. However, this analysis yielded less clear patterns. In the left hemisphere, we observed the medial part of the principal structural connectivity gradient being linked to the temporal pole and hence mostly limbic regions. Lateral thalamic portions were linked to a dispersed pattern of superior regions, while peaking in visual and dorsal attention networks. Noteworthy, the somatomotor network tended to not show a differential association between both anchors of the principal gradient, possibly suggesting more global and unspecific effects of covariance. Further, in contrast to structural and functional connectivity, we found that the pattern of structural covariance deviates between the left and right hemisphere, which may be linked to asymmetries in microstructure across individuals. In the cortex, previous studies have shown that there is a strong association between structural covariance and connectivity between areas[76–78]. This observation can be associated with the framework of the 'structural model', stating that cortical areas with a similar microstructure, in particular laminar differentiation, are also more likely to be structurally and functionally linked[79–82]. Extending on studies focusing on structural covariance in the cortex, structural covariance between the thalamus and cortex has been used to parcellate the thalamus in mice[54]. This work suggested that thalamocortical regions that were connected tend to structurally covary. At the same time, not all structurally covarying regions were connected, possibly pointing to indirect pathways for maturation and connectivity[54]. Translating this approach to humans, in the current work, we also found that structural connectivity was not paralleled by structural covariance in all regions. One possibility for explaining structural covariance could be the coactivation of physically connected regions and thereby activity-induced synaptogenesis in a coordinated fashion[52]. An alternative explanation might be that structural covariance arises from coordinated developmental processes[51] associated with transcriptomic similarity[83]. Of note, we found much stronger and differentiable patterns of structural covariance along the principal compared to the secondary gradient. Adding to the observation of our principal gradient resembling an axis shown to be related to genes that are relevant for thalamic development[43] and transcriptional profiling in mice[36], this supported the interpretation of the principal gradient being a more developmentally guided pattern.

## Limitations
To study the organization and connectivity of the human thalamus, the current study was based on in vivo MRI. Compared to *post-mortem* studies this comes with the advantage of straightforward data collection of multiple modalities but with the caveats of noise and lower spatial resolution. In contrast to tracer studies, we note that probabilistic tractography does not reflect connectivity at an axonal level but rather estimates larger fiber tracts and can lead to false positives. Inaccuracies can occur with sharply curved, closely neighboring, or poorly myelinated connections. Although, it has been reported that probabilistic tractography results correspond well to white matter anatomy[84]. Furthermore, we note that in this study only TC connectivity is considered, however, it is well-known that the thalamus is also strongly connected with the subcortex. To decode the gradients, we use the THOMAS atlas[56,57] to subdivide the thalamus into its constituent nuclei. However, it is important to note that the resolution of our data imposes limitations on the spatial precision of this subdivision, particularly for smaller nuclei, such as the Habenular nucleus and MGN. Therefore,

conclusions based on the atlas should be interpreted with caution. Utilizing 7 T data in future studies is necessary to provide more granular insights into thalamic organization[85]. Notably, the LGN is not included in our thalamic mask due to its extreme posteroventral peripheral location. Additionally, the present study focuses on group-level analysis, aiming to reveal generalizable organizational principles of the thalamus. While this approach provides valuable insights into group-level patterns, individual-level variability in thalamic structure remains unexplored in this work. Future research could address this limitation by incorporating subject-specific analyses to uncover how these gradients manifest at an individual level. Furthermore, exploring gradual differences in thalamic organization does not preclude the existence of thalamic subnuclei, which can be cytoarchitectonically and functionally delineated with sharp borders. However, we propose that clustering approaches might not account for all overlaps between connectivity profiles in the thalamus, and investigating gradual variations could help to understand thalamic organization principles. Though in the current work, we focused on structural gradients 1 and 2, these gradients only explained 46.42% (LH) of the variance. The first two spatial patterns in structural organization are readably interpretable informed by both our analysis and prior findings[36,43,45,47], and gradients explaining less variance become more difficult to give biological meaning and interpretation. Future research may benefit from examining multiple dimensions concurrently, akin to recent approaches linking cortical modes[86]. It should be noted, that the thalamus, being deep in the brain, is a relatively noisy region, and thus signal-to-noise ratio (SNR) must be considered as a potential confounder for our gradient measurements (see Supplementary Fig. 9 and Supplementary Table 4). To reduce this impact, we used group-level connectivity matrices, which help to average out individual differences. However, future research may benefit from exploring inter-individual variance, as it could hold biological relevance, though this lies beyond the scope of our current work.

In sum, we illustrated how the intrinsic organization of the human thalamus, as defined by TC white matter connections derived from DWI, mirrors thalamic microstructure and functional connectivity, and aligns with distributed structural and functional projections shaping cortical organization. Specifically, we could identify two axes of thalamic organization, which recapitulate differentiable thalamocortical structural and functional connections and may be rooted in differentiable neurobiological mechanisms of development and maturation. Future work incorporating multimodal high-resolution imaging and cognitive tasks may help to understand how the anatomy of the thalamus matures and shapes cortical structure, intrinsic function, and ultimately cognitive functional processes.

## Methods
### MRI data acquisition
For this study, we used the openly available multimodal MRI dataset for Microstructure-Informed Connectomics (MICA-MICs)[87,88]. This dataset was acquired from 50 healthy adults (23 women; 29.54 ± 5.62 years) and can be downloaded from the Canadian Open Neuroscience Platform's data portal (https://portal.conp.ca) and OSF. The complete cohort underwent a multimodal scanning protocol including high-resolution T1w, myelin-sensitive qT1 relaxometry, DWI, and rsfMRI at a field strength of 3 T. Scanning was conducted at the Brain Imaging Center of the Montreal Neurological Institute using a 3 T Siemens Magnetom Prisma-Fit scanner equipped with a 64-channel head coil. The original study was approved by the ethics committee of the Montreal Neurological Institute and Hospital. All participants provided written informed consent. All ethical regulations relevant to human research participants were followed. Exact acquisition protocols are described elsewhere[87]. In brief, they contained the following:

**T1w.** Using a 3D magnetization-prepared rapid gradient-echo sequence (MP-RAGE), two T1w images with identical parameters (0.8 mm isotropic voxels, matrix = 320 × 320, 224 sagittal slices, TR = 2300 ms, TE = 3.14 ms, TI = 900 ms, flip angle = 9°, iPAT = 2, partial Fourier = 6/8) were acquired.

**DWI**. For the acquisition of the multi-shell DWI data, a spin-echo echo-planar imaging sequence was used. This sequence consists of three shells with b-values 300, 700, and 2000 s/mm$^2$ and 10, 40, and 90 diffusion weighting directions per shell, respectively (1.6 mm isotropic voxels, TR = 3500 ms, TE = 64.40 ms, flip angle = 90°, refocusing flip angle = 180°, FOV = 224 × 224 mm$^2$, slice thickness = 1.6 mm, multi-band factor = 3, echo spacing = 0.76 ms, number of b0 images = 3). Additionally, b0 images in reverse phase encoding directions are provided for distortion correction of DWI scans.

**rsfMRI**. Resting-state fMRI images were acquired during a 7 min scan session using multiband accelerated 2D-BOLD echo-planar imaging (3 mm isotropic voxels, TR = 600 ms, TE = 30 ms, flip angle = 52°, FOV = 240 × 240 mm$^2$, slice thickness = 3 mm, mb factor = 6, echo spacing = 0.54 ms). For distortion correction of fMRI scans, two spin-echo images with reverse phase encoding (3 mm isotropic voxels, TR = 4029 ms, TE = 48 ms, flip angle = 90°, FOV = 240 × 240 mm$^2$, slice thickness = 3 mm, echo spacing = 0.54 ms, phase encoding = AP/PA, bandwidth = 2084 Hz/Px) were acquired. Participants HC001 to HC004 underwent slightly longer acquisition (800 time points) but for consistency, we use the same number of time points for all subjects (700 time points).

**qT1**. The qT1 relaxometry data were acquired using a 3D magnetization prepared 2 rapid acquisition gradient echoes sequence (MP2RAGE; 0.8 mm isotropic voxels, 240 sagittal slices, TR = 5000 ms, TE = 2.9 ms, TI 1 = 940 ms, T1 2 = 2830 ms, flip angle 1 = 4°, flip angle 2 = 5°, iPAT = 3, bandwidth = 270 Hz/px, echo spacing = 7.2 ms, partial Fourier = 6/8). To reduce sensitivity to B1 inhomogeneities and to optimize intra- and inter-subject reliability, two inversion images were combined for qT1 mapping.

## Preprocessing
For the preprocessing of MRI data, we used the specific modules of the containerized multimodal MRI processing tool *micapipe* (v. 0.1.2)[89] to ensure robustness and reproducibility. Details of the pipeline are described in Cruces et al. [89].

The below described modality-specific preprocessing steps depend on the output of the module for preprocessing T1w images including reorientation to LPI Orientation, linear alignment and averaging of T1w-scans, intensity nonuniformity correction (N4)[90], intensity normalization and a nonlinear registration to MNI152 using ANTS[91]. Building up on this, *micapipe* runs FreeSurfers' recon-all pipeline to segment the cortical surface from native T1w scans[92].

**DWI**. The used DWI preprocessing module comprises the alignment of multi-shell DWI scans via rigid-body registration, denoising using the Marchenko-Pastur PCA (MP-PCA) algorithm[93], Gibbs ringing correction[94], the correction of susceptibility-induced geometric distortions, eddy current-induced distortions, and head movements[95,96]. Further, a non-uniformity bias field correction was performed[90].

**rsfMRI**. The rsfMRI time series data were preprocessed using the adequate *micapipe* module. The first five volumes were dropped to guarantee magnetic field saturation. Images were reorientated (LPI), motion corrected by registering all time-point volumes to the mean volume, corrected for distortion using the main phase and reverse phase field maps, and denoised using an ICA-FIX classifier[97,98]. Further, motion spikes were removed using FSL and the average time series was used for registration to FreeSurfer space. The cortical time series were smoothed with a 10 mm Gaussian kernel and nodes were averaged defined by schaefer 200 parcellation scheme[99]. For the time series extraction of thalamic voxels, the preprocessed volumes in native space were warped to MNI152 standard space (isotropic resolution 2 mm) using the ANTS transformation parameters provided by *micapipe*.

**qT1**. QT1 is a proxy for gray matter myelin and provides an index for microstructure. The parameter refers to the T1 relaxation time in milliseconds, which is lower in fatty tissue compared to aqueous tissue[66]. Accordingly, note that qT1 reflects the amount of gray matter myelin in an inverse relationship. To map the individual qT1 scans to MNI152 2 mm standard space, for each subject the transformation between the uni_T1 map with removed background and the MNI152 1 mm reference image was computed using nonlinear registration. Next, the resulting transformation matrix was applied to the individual qT1 scan and data was downscaled to 2 mm isotropic resolution. The thalamic voxel-wise qT1 values were then extracted using a thalamus mask.

The qT1 intensity profiles per cortical parcel are provided by *micapipe*. Therefore, the module performs a registration from the native qT1 volume to FreeSurfer native space, followed by the construction of 14 equivolumetric surfaces from pial to white-matter boundary[100]. Based on these surfaces, depth-dependent intracortical intensity profiles at each vertex of the native surface mesh are generated and averaged within each of the cortical parcels (schaefer 200 parcellation)[99]. For each subject, this resulted in a matrix containing qT1 values and with the dimensions L x N, where L is the number of equivolumetric compartments and N the number of cortical parcels. For the analysis, the pial and gray matter/white matter surface, as well as the medial wall were excluded, resulting in a 12 x 200 matrix per subject, which was then averaged across the 12 compartments.

## Thalamic mask
The binary thalamic mask in MNI152 standard space (2 mm isotropic resolution) was created based on the Harvard-Oxford subcortical atlas, integrated in FSL[101], for each hemisphere, respectively. To incorporate only thalamic voxels in the subsequent analysis and mitigate signal bleeding i.e., from the third ventricle, the thalamic masks underwent a refinement via manual thresholding (left: dilation of 19%, right: dilation of 20%).

## TC structural connectivity and gradient decomposition
**Voxel-wise distribution estimates of diffusion parameters**. To estimate the voxel-wise distribution of fiber orientations in the preprocessed diffusion weighted images, we used FSL's function *bedpostX* (Bayesian Estimation of Diffusion Parameters)[102]. The multi-shell extension of the ball- and sticks model was utilized, with a maximum estimate of three fiber orientations within each voxel[103].

**TC probabilistic tractography**. On a subject level, probabilistic tractography between each voxel in the thalamic seed mask and each parcel of the cortical termination mask (schaefer 200 parcellation[60,99]) was computed using FSL's *probtrackx2* (probabilistic tracking with crossing fiber)[104]. Tractography was computed independently in the left and right hemispheres, in line with previous studies and due to predominant ipsilateral projections[19,26,43]. Each seed voxel was sampled 5000 times with a curvature threshold of 0.2 and step length of 0.5 mm. The path probability maps were corrected for the length of the pathways. Considering that seed- and target masks are provided in MNI152 standard space, it was additionally required to pass the transformation matrices between native diffusion and MNI standard space. For each subject, we constructed individualized transformations by within-subject registration of the brain extracted b0 image to the native structural scans using FSL's *FLIRT* (6 degrees of freedom) concatenated to the nonlinear between-subject registration of native structural scans to MNI standard template in 2 mm resolution using a combination of *FLIRT* and *FNIRT* (12 degrees of freedom).

**TC structural connectivity matrix computation**. Using the output of *probtrackx2*, for both hemispheres a structural connectivity matrix (thalamic seed voxels x cortical parcels) was computed, containing the numbers of streamlines between seed voxels and target parcels per individual. We averaged the individual connectivity matrices across subjects to form a group-level structural connectivity matrix. The group-

level structural connectivity matrix was normalized column-wise by dividing all values of each column by the maximum of this column.

**TC structural connectivity gradients.** To uncover the intrathalamic organization based on the TC structural connectome, we employed nonlinear dimensionality reduction. This results in gradient components representing low dimensionally the variation in the connectivity data in a gradual manner. To this end, several analysis steps were performed using the Python toolbox *BrainSpace* (v. 0.1.3)[105]. The TC group-level structural connectivity matrix was thresholded at the 75th percentile, in line with previous work[67], and converted into a non-negative squared affinity matrix using a normalized angle similarity kernel. Next, we applied diffusion map embedding, a non-linear dimensionality reduction technique that belongs to the family of graph Laplacians[106] to derive a low-dimensional embedding from the high-dimensional affinity matrix. In this manifold space are nodes with similar connectivity profiles embedded closer together, whereas nodes with distinct connectivity patterns are located further apart. The algorithm is controlled by the $\alpha$ parameter, which determines the density of sampling points on a manifold (where 0 to 1 = maximal to no influence). Following recommendations based on previous work[38,41,107], we set $\alpha$ to 0.5 as that will preserve large scale relations between data points in the embedded space and has been suggested to be relatively robust to noise. After assessing the amount of explained variance for each gradient, we mapped the resulting first four gradients ($G1_{sc}$ - $G4_{sc}$) onto the thalamic mask. Note that in the following analysis, we focus on the principal ($G1_{sc}$) and secondary gradient ($G2_{sc}$).

**Contextualization with THOMAS atlas**
To identify the relationship between discrete defined anatomical thalamic subnuclei and TC structural connectivity gradients, we used the parcellation of the THOMAS (Thalamus optimized multi atlas segmentation) atlas in MNI152 space including 11 nuclei[56,57] (https://doi.org/10.5281/zenodo.5499504). First, we spanned a 2D space framed by $G1_{sc}$ and $G2_{sc}$ and categorized the thalamic voxels according to the nucleus they are part of. Second, for $G1_{sc}$ and $G2_{sc}$, we extracted per nucleus the corresponding gradient loadings and sorted the nuclei in an ascending order based on the median of the corresponding gradient loadings.

**Multimodal intrathalamic spatial organization**
**Intrathalamic qT1 map.** By using the thalamic mask, we extracted the thalamic qT1 values at the subject level. Subsequently, we computed the voxel-wise qT1 average across subjects, resulting in a group-level qT1 map of the thalamus.

**Cell type map based on gene expression levels of Parvalbumin and Calbindin.** To approximate the spatial distribution of cell types in the thalamus, we used a relative difference map of core and matrix cells provided by Müller et al.[59] and publicly available (https://github.com/macshine/corematrix). The map is derived from mRNA level estimates of genes that express the calcium-binding proteins Parvalbumin (PV) and Calbindin ($CB_1$) supplied by the Allen Human Brain Atlas[108]. PV and $CB_1$ have been shown as adequate markers for distinguishing between core- and matrix cells in the thalamus. Considering the sample size on which the data is grounded, we opted to exclusively use the map of the left hemisphere map (6 donors), while excluding the right hemisphere (2 donors). We determined the intersection of this map and our thalamic mask and referred to it as core-matrix map.

**Functional connectivity gradients.** For each subject, we correlated (Pearson) the intrahemispheric timeseries of thalamic voxels and cortical parcels to create a TC functional connectivity matrix per hemisphere (thalamic voxels x cortical parcels). Rows were Fisher z-transformed. By averaging across subjects, the group-level TC functional connectivity matrix was calculated. In line with previous work[38,107], this matrix was

thresholded at the 90th percentile and used as input to estimate the functional connectivity gradients using diffusion map embedding, analog to gradient computation of structural connectivity described above. $G1_{fc}$ and $G2_{fc}$ were mapped onto the thalamic mask.

**Association between TC structural connectivity gradients and intrathalamic features.** Next, we explored the association between the spatial thalamic organization based on TC structural connectivity and (a) the distribution of thalamic qT1, that is used as a proxy for gray matter myelin, (b) the relative amount of core and matrix cells, and (c) functional connectivity gradients. Therefore, we spanned a 2D space framed by structural connectivity $G1_{sc}$ and $G2_{sc}$ and color-coded the data points based on (a) the group-level qT1 value, (b) the relative difference between Calbindin and Parvalbumin levels, and (c) the principal and secondary functional connectivity gradient loadings. To further quantify the relationship, we calculated the Pearson correlation of the structural connectivity gradients with (a) the group-level qT1 map, (b) the cell type map estimating the relative amount of core- and matrix cells, and (c) the principal and secondary functional connectivity gradients. All analyses were corrected for spatial autocorrelation (SA) using the variogram approach implemented in the brainsmash toolbox[58]. In addition, we show the correlation between all ten gradient components and the intrathalamic features (Supplementary Table 1).

**Multimodal TC spatial organization**
**Mapping TC structural connectivity gradients on the cortex.** To project the low-dimensional thalamic organization of TC structural connectivity onto the cortex, we calculated the Pearson correlation of $G1_{sc}$ with each column (representing the parcels) of the group-level TC structural connectivity matrix and projected the resulting Pearson correlation coefficient per parcel onto the cortex surface. The analogous procedure was performed for $G2_{sc}$.

**Mapping the association between TC structural connectivity gradients and functional connectivity on the cortex.** To access the link between the thalamic organization based on TC structural connectivity and TC functional connectivity, we calculated the Pearson correlation of $G1_{sc}$ with each column (representing the parcels) of the group-level TC functional connectivity matrix and projected the resulting Pearson correlation coefficient per parcel onto the cortex surface. The same was applied to $G2_{sc}$.

**TC structural covariance.** We further aimed to investigate whether the TC structural connectivity relates to TC structural covariance. Therefore, we computed the intrahemispheric TC structural covariance matrix by Pearson correlating qT1 values of thalamic voxels with qT1 values of cortical parcels across subjects. This resulted in a TC structural covariance matrix per hemisphere (thalamic voxels x cortical parcels).

**Mapping the association between TC structural connectivity gradients and TC structural covariance on the cortex.** To probe the link between the thalamic organization based on structural connectivity and TC structural covariance, we computed for each parcel the Pearson correlation coefficient by correlating $G1_{sc}$ with each column (representing the parcels) of the group-level TC structural covariance matrix and projected the results onto the cortical surface. The analogous procedure was applied to $G2_{sc}$.

**Decoding with functional networks.** For each modality, the cortical projection patterns were decoded using functional network communities (Visual, Somatomotor, DAN, VAN, Limbic, FPN and DMN)[60].

**Statistics and reproducibility**
The data sample (MICA-MICs) included 50 individuals with a mean age ± SD of 29.54 ± 5.62 years. We examined spatial correlations between

different modalities in the thalamus (LH: 1068 voxels, RH: 1029 voxels) using Pearson correlation tests, and corrected for spatial autocorrelation using variograms[58]. Custom code is provided to ensure the reproducibility of our study.

## Reporting summary

Further information on research design is available in the Nature Portfolio Reporting Summary linked to this article.

## Data availability

The dataset used in this study is openly available and can be downloaded from: CONP Portal (https://portal.conp.ca/dataset?id=projects/mica-mics) and OSF (https://osf.io/j532r/ with the identifier https://doi.org/10.17605/OSF.IO/J532R)[87,88]. As a basis for our thalamus mask, we used the openly accessible Harvard-Oxford subcortical atlas, integrated in FSL. The template of the THOMAS atlas, which was used to identify thalamic nuclei, can be requested here: https://doi.org/10.5281/zenodo.5499504[57]. The core-matrix difference map can be downloaded from: https://github.com/macshine/corematrix[59]. For supplementary analysis, we compared our structural connectivity gradients with those from Oldham and Ball[43]; their source data is available here: https://doi.org/10.1038/s41467-023-41722-8. Data generated in this study, such as group-level structural connectivity matrix, group-level functional connectivity matrix, group-level structural covariance, and group-level qT1 map have been deposited on Github (https://github.com/CNG-LAB/cngopen/tree/main/thalamic_gradients). Source data behind the main figures can be found in Supplementary Data 1.

## Code availability

For data preprocessing and tractography, we used the openly available processing pipeline *micapipe* (v. 0.1.2; https://micapipe.readthedocs.io/) and FSL. Custom code generated for analysis and plotting is available at a Github repository (https://github.com/CNG-LAB/cngopen/tree/main/thalamic_gradients). Our analysis code makes use of open software: The gradient computation and cortical surface visualization were carried out using BrainSpace (v. 0.1.3; https://brainspace.readthedocs.io/en/latest/). Statistical analysis was done using scipy (v. 1.7.3) and numpy (v. 1.21.2). For spatial autocorrelation correction, the brainsmash toolbox was used (v. 0.11.0; https://brainsmash.readthedocs.io/en/latest/). For further analysis and visualization, we made use of: pandas (v. 0.2.0), nibabel (v. 3.2.1), nilearn (v. 0.8.1), seaborn (v. 0.11.2), ptitprince (v. 0.2.5), and matplotlib (v. 3.4.3).

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

## Acknowledgements
We want to express their gratitude to the various contributors to the MICA-MICs dataset and for openly sharing the dataset with the neuroscientific community. S.L.V., A.J., A.S., B.W. and L.H.S. were supported by the Max Planck Society through the Otto Hahn Award. S.L.V., B.C.B. and A.S. are furthermore funded by the Helmholtz International BigBrain Analytics and Learning Laboratory (HIBALL), supported by the Helmholtz Association's Initiative and Networking Fund and the Healthy Brains, Healthy Lives initiative at McGill University. BCB acknowledges research support from the National Science and Engineering Research Council of Canada (NSERC Discovery-1304413), Canadian Institutes of Health Research (FDN-154298, PJT-174995, PJT-191853), SickKids Foundation (NI17-039), BrainCanada, FRQ-S, and the Tier-2 Canada Research Chairs program. J.R. was supported by a fellowship from the Canadian Institutes of Health Research. MDH was funded by the German Federal Ministry of Education and Research (BMBF) and the Max Planck Society.

## Author contributions
A.J.: Conceptualized the study and manuscript; wrote custom computer code; performed analyses; interpretation and visualization of results, wrote the manuscript; incorporated coauthor revisions of manuscript, incorporated reviewers' suggestions. M.D.H.: Provided assistance in conceptualizing the study, interpretation of results, edited, and reviewed the manuscript. H.L.S.: Provided assistance in conceptualizing the study, in interpretation of results, edited and reviewed the manuscript. A.S.: Provided assistance with methodology and analysis, edited and reviewed the manuscript. S.B.: Technical assistance. B.W.: Provided assistance with custom code, reviewed the manuscript. J.R.: Provided the dataset and assistance with preprocessing, edited, and reviewed the manuscript. B.C.B.: Edited and reviewed the manuscript, provided supervision. S.L.V.: Conceptualized the study and manuscript; assistance in interpretation of results, edited and revised comments on the manuscript and computer code (multiple occasions); provided supervision; provided funding for the project.

## Funding

## Competing interests

The authors declare no competing interests.
