## [Transparent Peer Review file · Communications Biology]

A Multimodal Characterization of Low-Dimensional Thalamocortical Structural Connectivity Patterns

Corresponding Author: Ms Alexandra John

Version 0:

Reviewer comments:

Reviewer #1

(Remarks to the Author)

This study derived the structural connectivity gradients of the thalamocortical connections in 50 healthy young adults and compared the spatial patterns of the thalamus gradients with several micro- and macro-scale neural properties of the thalamus as well as the corresponding cortical projections.

This is an interesting study with a rich set of analyses. The manuscript is also well-written with proper discussions. Here are my comments:

- i) It is unclear why this study focuses on the structural connectivity but not the functional connectivity gradients. Apparently, this whole study can be repeated using the functional connectivity gradients as the primary target and compare the FC gradients with SC and other micro- and macro-scale properties. A strong rationale is needed here.
 - ii) Although I am not entirely familiar with the literature of thalamus gradients, it would be a surprise that the SC gradients of the thalamus has not been studied previously. It would be important to clarify the extent to which the FC/SC gradients of the thalamus have been mapped in the literature and the extent to which this work builds upon previous studies. The novelty of studying SC gradients of the thalamus needs to be clarified.
 - iii) The thalamus has been studied extensively in the literature. Although this study involves a lot of comparisons that may have not been done in a single previous study, most of the comparisons and analyses are spatial correlations of pairs of group-level maps. Although it reveals some interesting spatially matched patterns across modalities, the coherence of the findings can be further improved. It seems that the core conclusion is that the thalamus is heterogenous, which is already known by the field, but how do the findings and conclusions of this study advance our understanding of the thalamus and its role in brain organization and function?
 - iv) Given that this is a group-level analysis, it is important to describe the sample size in the Results section. The current manuscript is a bit misleading because in the Intro and Results sections, it is described that the benefit of using MRI is because of the large sample sizes; however, the sample size is only 50, which is much smaller than most contemporary neuroimaging studies, for example, the human connectome project with 1000 individuals. It is unclear why this study used a cohort with only 50 individuals rather than using the HCP dataset, which is open access and has all the imaging modalities they analysed. It is therefore unclear whether the findings reported here are reproducible and generalizable to a wider population and in a larger sample.
- Minor point: What are the data points presented in Figure 1F and other 2D plots of G1 and G2 in other figures? Are they thalamus voxels?

Reviewer #2

(Remarks to the Author)

1. The analysis is well-executed, utilizing advanced preprocessing and post-processing techniques to depict the low-dimensional connectivity patterns between the thalamus and cortex, i.e., gradients. However, the results remain speculative due to the partial selection of gradients and the broader association of both gradients with the thalamus nuclei. It is difficult to comprehend the results with respect to the two modes of organization in the depicted gradients (Figure 1) and associated results and discussion.
2. Given that the thalamus is significantly affected by a lack of SNR/tSNR, it is essential to assess the quality of the results. Please provide the thalamus SNR values in violin plots for all subjects for the qT1, T1 images, and tSNR maps, as well as of

for the DWI and rsfMRI data.

3. It is also unclear why only two gradients, explaining 46% of the variance (Figure 1C), were included, without accounting for the rest to illustrate the intricacies between the thalamus and cortex.

4. Figure 1F, which decodes based on the THOMAS atlas, shows a diffuse and sparse association of nuclei in both gradients, resulting in a non-specific nuclei association, contrary to the discussed axis organization. It is advisable to discuss the findings in a broader context while critically examining the observations. In the current stage it is difficult to comprehend the axis organization and its biological meaning as well the differences in the structural, functional and structural covariance.

5. The thalamus comprises smaller nuclei, and fully understanding their representation in a gradient ideally requires a much higher resolution fMRI scan than the 3mm isotropic resolution used in this analysis.

6. If the THOMAS thalamus segmentation method was applied to individual subjects, the nuclei volumes would reveal inter-individual differences. Therefore, it is important to analyze these differences in addition to the group-level analysis.

Version 1:

Reviewer comments:

Reviewer #1

(Remarks to the Author)

My previous comments have been adequately addressed. I have no further comments.

Reviewer #2

(Remarks to the Author)

The work presents itself as well-organized, with explanations for the methodologies used. The conclusions drawn appear to be supported by the data and analysis presented. Additionally, the paper is written with detail to allow for reproducibility by other researchers in the field.

At this time, I have no further comments or concerns to raise.

COMMSBIO-24-0793

Title: “A Multimodal Characterization of Low-Dimensional Thalamocortical Structural Connectivity Patterns”

Response Letter to Referees

We would like to thank the Editors and Reviewers for their positive evaluations, constructive comments, and for the opportunity to submit a revised manuscript. We feel that the comments and suggestions have greatly improved our work. In this response letter, we outline the steps taken to address the suggestions of the Reviewers in a point-by-point fashion below and highlight the corresponding changes in the manuscript in yellow.

Reviewer #1 (Remarks to the Author):

This study derived the structural connectivity gradients of the thalamocortical connections in 50 healthy young adults and compared the spatial patterns of the thalamus gradients with several micro- and macro-scale neural properties of the thalamus as well as the corresponding cortical projections.

This is an interesting study with a rich set of analyses. The manuscript is also well-written with proper discussions. Here are my comments:

Q 1: It is unclear why this study focuses on the structural connectivity but not the functional connectivity gradients. Apparently, this whole study can be repeated using the functional connectivity gradients as the primary target and compare the FC gradients with SC and other micro- and macro-scale properties. A strong rationale is needed here.

A 1: We are happy to further clarify the rationale of our study for focusing on white matter connectivity of the thalamus. Our study aims to understand the characteristics of the low-dimensional thalamic organization by taking into account multiple modalities such as structural connectivity, myelin and core-matrix cell distribution, functional connectivity and structural covariance. The thalamus can be characterized based on various modalities, and previous studies have begun to investigate thalamocortical functional connectivity using the gradient approach in adults (Yang et al., 2020) and normative development (Park et al., 2024). Likewise, structural connectivity gradients have been investigated (Oldham and Ball, 2023; see also Q2). However, our study seeks to extend these findings by employing a multimodal framework, starting with structural connectivity as the primary focus. While we appreciate the Reviewers' suggestion to prioritize functional connectivity for comparisons, our approach is grounded in the following rationale:

- The thalamus is structurally connected with the cortex and it is known from animal studies and human DWI studies that different thalamic subareas show different structural connectivity patterns (e.g., cortical target area; Halassa and Sherman, 2019; Sherman, 2012; Behrens et al., 2003). These connections are formed from early development onwards and are biologically grounded in a heterogeneous distribution of structural entities such as different thalamic cell types (e.g., Parvalbumin+ and Calbindin+; Jones, 1998). Our aim was to understand the complex thalamic organization from a structural point of view and to bring these patterns in context with other structural measures such as local myelin distribution, core-matrix cell distribution and structural covariance. Indeed, to characterize our gradients, we also aimed to investigate whether there is an overlap between our structural connectivity patterns and functional connectivity patterns, although the focus in this study is clearly on the structural site.
- In our study, our goal was to understand the organization of the thalamus based on its connection to the cortex from a multimodal point of view. Hence, structural connectivity, which shows the direct physical connections between brain regions through white matter tracts, was well-suited for our study. In contrast, functional connectivity potentially also captures indirect connections that might reflect polysynaptic pathways between regions. While diffusion weighted imaging has its limitations (such as potential misleading tracts, high susceptibility to motion artifacts), studies like Behrens et al. (2003) have shown that thalamocortical probabilistic tractography results can be compared favorably with tracer studies from animals. This supports its use as a reliable basis for analyzing structural connectivity and enhances our understanding of the monosynaptic connectional arborization within the thalamocortical system.
- Lastly, gradual variations in features such as microstructure along the thalamic axis may be based on the molecular gradients of morphogens and transcription factors. This assumption is supported by a study in mice conducted by Phillips et al. (2019), which demonstrates a gradual organization of gene expression associated with cellular variation, electrophysiological properties, and axonal morphology. This axis is related to specific patterns of axonal projections. Therefore, we argue that in humans, structural connectivity patterns may reflect these microstructural variations and serve as a good starting point for our analysis.
- Though a post-hoc argument arising from the revision process, we demonstrate in Q3 that structural connectivity serves as a more robust measure of general organizational principles. Specifically, it exhibits less variance between individual and group-level gradients compared to functional connectivity. This finding supports that using structural connectivity to investigate general organizational patterns might be a better measure from a data-driven perspective.

We have now included a summary of this rationale in the last paragraph of our introduction (line 92):

“The heterogeneous white matter connections between the thalamus and cortex, which form early in development and are biologically based on a heterogeneous distribution of structural entities such as different thalamic cell types, are suitable as a direct measure to study thalamic organization (Behrens et al., 2003; Phillips et al., 2019; Oldham and Ball, 2022; Zheng et al., 2023; Kim et al., 2023). In the current study, we therefore explored how the internal organization of the human thalamus based on its structural connections to the cortex corresponds with the distribution of thalamic microstructural features, as well as TC functional connectivity and structural covariance. [...]”

Moreover, for the interested reader, we have added an analysis showing the link between the functional connectivity gradients and the local microstructural measures to the Supplementaries with the following changes:

Results (line 195):

“Additionally, in a supplementary analysis, we explored the association between functional connectivity gradients and the group-level qT1 and core-matrix maps, revealing a correlation between $G2_{fc}$ and the core-matrix map (LH: $r = 0.568$, $p_{SA} = 0.01$) (Supplementary Table 2).”

Discussion (line 360):

“Notably, the second functional gradient was also correlated with the cell type distribution, suggesting that the core and matrix cells serve as a basis for both structural and functional organizational patterns.”

Supplementary Analysis (line 763):

“Association between functional connectivity gradients and thalamic microstructure

To additionally investigate associations between thalamic functional connectivity gradients and thalamic microstructure, we calculated Pearson correlation between the first two functional connectivity gradients and the group-level qT1 map and core-matrix map.”

Supplementaries:

	group-level qT1 map	core-matrix difference map
functional connectivity gradient 1	LH: $r = -0.041$, $p_{SA} = 0.94$ RH: $r = -0.129$, $p_{SA} = 0.76$	LH: $r = -0.211$, $p_{SA} = 0.76$
functional connectivity gradient 2	LH: $r = 0.085$, $p_{SA} = 0.84$ RH: $r = 0.069$, $p_{SA} = 0.89$	LH: $r = 0.568$, $p_{SA} = 0.01$

Supplementary Table 2: Association between TC functional connectivity gradients and thalamic microstructure. Pearson correlation between maps corrected for spatial autocorrelation (p_{SA}). Note that for the core-matrix map, only results of the left hemisphere are reported due to the small sample size on which the right core-matrix map is based. Abbreviations: LH: left hemisphere, RH: right hemisphere.

Q 2: Although I am not entirely familiar with the literature of thalamus gradients, it would be a surprise that the SC gradients of the thalamus has not been studied previously. It would be important to clarify the extent to which the FC/SC gradients of the thalamus have been mapped in the literature and the extent to which this work builds upon previous studies. The novelty of studying SC gradients of the thalamus needs to be clarified.

A 2: Thank you for raising this point about the foundation and novelty of our study. While gradients in the cortex have been extensively researched, in particular the relationship between structure and function of gradients (Bernhardt et al., 2022; Margulies et al., 2016; Smallwood et al., 2021; Valk et al., 2020a; Paquola et al., 2019; Huntenburg et al., 2018), there has been comparatively less focus on subcortical gradients. In the following, we report existing literature on structural (Oldham and Ball, 2023; Howell et al., 2024; Zheng et al., 2023) and functional connectivity thalamic gradients (Yang et al., 2020; Park et al., 2024) and how our study extends on them.

Our study aimed to extend this body of work by characterizing thalamic gradients from a multimodal thalamo-centric perspective to further understand the complex thalamic organization. We integrate structural connectivity, qT1 as local myelin measure, the core-matrix framework, functional connectivity and structural covariance to provide a more comprehensive understanding of thalamic gradients. By doing so, we build upon existing research and offer new insights into the structural connectivity gradients of the thalamus.

Our work extends work from Oldham and Ball (2023), where thalamic gradients were computed based on structural connectivity data from 74 subjects of HCP in joint decomposition with post-mortem gene expression data. The authors found a medial-lateral axis similar to ours, and also present in mice. Based on gene expression data, they found this axis related to neuronal subtype distribution, development and disease. While this study linked the gradients to gene expression data, we add to this by characterizing the structural connectivity gradients (calculated without gene expression data) to thalamic (qT1, rsfMRI gradients) and thalamocortical interrelations (functional connectivity and structural covariance) measured by MRI and the core-matrix maps, derived from gene expression data.

More recently, Howell et al. (2024) investigated the spatial extent of focal and diffuse thalamocortical connectivity patterns using euclidean distance measures. Therefore, they computed probabilistic tractography in humans (N = 828) and macaque monkeys (N=6), where connectivity profiles have been shown to vary along the cortical hierarchy (sensory-to-association). While their approach aims to understand the spatial distribution of thalamic focal and diffuse connectivity patterns, they mainly take a cortico-centric view by comparing their patterns to cortical maps (e.g. cortical T1W/T2W, cortical functional gradients). In contrast, our study reports structural connectivity gradients and their association to intrathalamic microstructural patterns, thalamocortical functional connectivity and structural covariance, which characterizes the internal thalamic organization from a multimodal and thalamocortical perspective.

Further, Yang et al. (2020) investigated thalamic gradients based on functional thalamocortical connectivity. They found a principal medial-lateral functional gradient associated with thalamic grey matter volume, and a second anterior-posterior axis corresponding to functional networks. Thalamocortical functional connectivity gradients have also been studied to investigate the role of the thalamus in development from infancy to young adulthood (Park et al., 2024). They found that the role of the thalamus may play a role in the decoupling externally- and internally-oriented functional processes in development. Here again the perspective is more cortico-centric, e.g. the focus was the influence of the thalamus on cortical development, but not internal thalamic organization as we focus on in our study.

We have revised and expanded the introductory paragraph to provide a more comprehensive review of studies on thalamic gradients, which form the foundation of our work, and to highlight the novel contributions of our study (line 67, 71).

[...] In line with this, recent advances of studying brain organization have shifted their focus on revealing spatially graded changes of neurobiological properties across the brain, in addition to the traditional approaches of defining discrete brain regions (Bernhardt et al., 2022; Margulies et al., 2016; Smallwood et al., 2021; Valk et al., 2020a; Paquola et al., 2019; Huntenburg et al., 2018). These continuous axes of spatial variation are referred to as 'gradients'. While this approach has mainly been applied to understand macroscale cortical organization, recent work uncovered transitional axes that help explain organizational patterns of the human thalamus (Oldham and Ball, 2022; Howell et al., 2024; Zheng et al., 2023; Yang et al., 2020; Park et al., 2024). These gradients, derived from thalamocortical structural and functional connectivity, might arise from smooth transition at the microscale level (Phillips et al., 2019; Roy et al., 2022). Based on the joint analysis of TC structural connectivity and gene expression data, a phylogenetically conserved medial-to-lateral axis has been reported that captured transitions in cell type variations and suggested a link to development and disease (Oldham and Ball, 2023). In addition, thalamic gradients derived from functional connectivity between the thalamus and cortex have been reported to follow a principal medial-to-lateral axis that was associated with thalamic grey matter volume, and a secondary anterior-to-posterior axis corresponding to functional networks (Yang et al., 2020). Recently, functional thalamocortical gradients have been used to investigate the thalamic

role on cortical organization in development (Park et al., 2024). From a cortico-centric perspective, it has been shown that the thalamic axis, reflecting variations in the spatial extent of corticothalamic structural connections, is linked to the sensory-association cortical hierarchy (Howell et al., 2024). Taking a thalamo-centric perspective and drawing lines between the thalamus' internal organizational patterns and its interrelation to the cortex across different modalities, may provide further valuable insights into this complex structure."

Q 3: The thalamus has been studied extensively in the literature. Although this study involves a lot of comparisons that may have not been done in a single previous study, most of the comparisons and analyses are spatial correlations of pairs of group-level maps. Although it reveals some interesting spatially matched patterns across modalities, the coherence of the findings can be further improved. It seems that the core conclusion is that the thalamus is heterogeneous, which is already known by the field, but how do the findings and conclusions of this study advance our understanding of the thalamus and its role in brain organization and function?

A 3: We thank the Reviewer for acknowledging the multimodal insights provided by our work. By integrating different modalities, with structural connectivity as a foundation, we identified two main organizational axes in the thalamus, differentiating microstructure and cell types.

At the same time, we also find, as others have done before us, the thalamus is a complex, structurally and functionally heterogeneous brain structure. It has been categorized at different scales and modalities, from immunochemistry and histology to MRI contrast-based parcellations and macroscale connectivity, with each approach adding insights to our understanding but at the same time facing specific challenges. For instance, the renowned Morel atlas which is often seen as ground truth includes details at the microscale, however was derived from a small sample of five subjects (Morel et al., 1997). MRI studies at the other hand resulted in different parcellation schemes from various methods (i.e., Behrens et al., 2003; Iglesias et al., 2018; Najdenovska et al., 2018). While our study does not fully resolve the complexities of thalamic organization, it does push the boundaries of current knowledge and contributes significantly to the existing literature by using a gradient decomposition as an alternative approach to clustering. This method offers the advantage of revealing the relationships between areas independent of their position, indicating how similar structural connectivity patterns are. Notably, we observed coherence in the principal patterns of thalamocortical connectivity and their associations with microstructure and functional connectivity, putting forward novel hypotheses on the developmental axes within the thalamus. Additionally, our study goes beyond focusing solely on the thalamus but also considers its relationship with the cortex, from the perspective of the thalamus, providing a more comprehensive view of brain organization. We believe that advancing our understanding of the thalamus requires integrating multiple perspectives, and our multimodal approach offers a novel contribution by synthesizing these diverse features.

We added the following paragraph into the discussion (line 293):

“We acknowledge the diverse methods used to study the thalamus, from immunochemistry and histology to MRI-based parcellations, each contributing valuable insights while facing unique challenges, such as obtaining sufficient high-quality specimens for detailed histological analysis and varying modality-specific MRI parcellation schemes. Our research uses a gradient decomposition approach, which reveals relationships between thalamic areas independent of their spatial location, highlighting similarities in structural connectivity patterns and its coherence with microstructure, and functional connectivity. Importantly, we also expand the focus to examine the thalamus in relation to the cortex, offering a broader perspective on brain organization. Our multimodal approach synthesizes different methodologies, advancing the understanding of thalamic organization in a more integrated and comprehensive way.”

Q 4: Given that this is a group-level analysis, it is important to describe the sample size in the Results section. The current manuscript is a bit misleading because in the Intro and Results sections, it is described that the benefit of using MRI is because of the large sample sizes; however, the sample size is only 50, which is much smaller than most contemporary neuroimaging studies, for example, the human connectome project with 1000 individuals. It is unclear why this study used a cohort with only 50 individuals rather than using the HCP dataset, which is open access and has all the imaging modalities they analyzed. It is therefore unclear whether the findings reported here are reproducible and generalizable to a wider population and in a larger sample.

A 4: We thank the Reviewer for highlighting this important point. To ensure clarity regarding the sample size in our manuscript, we have stated the sample size in the Results section (lines 113, 147, and 251), as well as in the legend of Fig 1, 2, and 3. Further, to avoid any misunderstandings, we have also rephrased and extended our argument for MRI data in the Introduction and Discussion, highlighting its possibility to acquire functional and structural data in vivo of multiple modalities.

Introduction (line 34) *“Compared to post-mortem studies, non-invasive neuroimaging using magnetic resonance imaging (MRI) provides the opportunity to acquire in vivo functional and structural data to study the thalamic organization and its relationship to the cortex in multiple modalities.”*

Discussion (line 489): *“To study the organization and connectivity of the human thalamus, the current study was based on in vivo MRI. Compared to post-mortem studies this comes with the advantage of straightforward data collection of multiple modalities but with the caveats of noise and lower spatial resolution.”*

On top of that, we acknowledge that the MICA-MICs dataset, with a sample size of 50, is smaller compared to large open datasets like the Human Connectome Project (HCP) which includes > 1000 individuals. However, the MICA-MICs dataset offers specific advantages that align with the goals of our study and supports our choice:

Our study aims to investigate structural gradients from a multimodal perspective, necessitating a dataset with multiple high-quality modalities. MICA-MICs, an open dataset designed for multiscale neuroscience, satisfies this criteria including multishell diffusion measures, quantitative T1 (qT1), and resting-state functional data. Given that two of our analyses rely on detailed microstructural measures (the intrathalamic microstructural map and structural covariance), the inclusion of qT1 as a high-quality proxy for grey matter myelin is beneficial for our study. Complementary to other widely-used modalities reflecting microstructure, such as T1w/T2w in HCP, quantitative T1 offers a physical measure particularly sensitive to myelin where shorter T1 relaxation times reflect higher myelin content (Paquola and Hong, 2023; Weiskopf et al., 2021; Sandrone et al., 2023). This specificity of the MICA-MICs dataset aligns with the objectives of our study, providing the necessary data quality to support our findings.

While the MICA-MICs dataset fulfills these criteria, it indeed comes with the downside of a smaller sample size. To ensure internal consistency, we added a supplementary analysis showing the variance between the group level gradient and gradients on the subject level. The same we show for intrathalamic microstructure and resting state functional connectivity. While there is not much variance between the individual structural connectivity gradients and the group-level gradient as well as the individual level qT1 and the group-level qT1 map, we see that there is variance in the functional connectivity gradients. (This also further justifies our decision to use structural connectivity as the starting point for our analysis, from both a conceptual and data-driven perspective (see Q 1).)

Methods (line 716): *“Within-modality correlation between the individual- and the group-level maps*

To investigate consistency, we additionally calculated structural connectivity gradients, functional connectivity gradients and qT1 maps at the individual level. Gradients were procrustes aligned to the group-level maps. Individual maps were then Pearson correlated with the corresponding group-level maps.”

Results (line 185): *“To evaluate consistency, we present the Pearson correlation (r) values between individual structural connectivity, qT1, and functional connectivity maps and their respective group-level maps. The findings indicate high consistency for the principal structural connectivity gradient and qT1, while functional connectivity gradients show the most variation across individuals (Supplementary Figure 5).”*

Supplementary Figure 5: Correlation between the individual- and corresponding group-level maps. Violin plot showing the r values of correlation (Pearson) between the group-level maps with the individual-level maps of each subject ($N = 50$) of structural connectivity gradients, functional connectivity gradients and qT1 maps. Abbreviations: *sc*: structural connectivity, *fc*: functional connectivity, *lh*: left hemisphere, *rh*: right hemisphere.

To further test generalizability of our results, we compared our gradients to those reported by Oldham and Ball (2023), where they used a combination of transcriptomic and structural connectivity data (76 subjects from HCP young adult sample) to generate thalamic gradients. If we would observe similar gradients in these independent samples this would argue for the generalizability of our observations.

Overall, we find that both our structural connectivity G1 and G2 had a spatial association with the main gradient reported in Oldham and Ball (2023) ($G1_{sc}$: $r = 0.478$, $p_{SA} = 0.037$; $G2_{sc}$: $r = -0.701$, $p_{SA} = 0.039$). When investigating the relation between our gradients and the gradients based on structure and gene expression separately, we observed a clear differentiation, with our $G1_{sc}$ tending to relate to genes ($r = 0.423$, $p_{SA} = 0.054$) but not structural connectivity ($r = -0.061$, $p_{SA} = 0.778$), yet our $G2_{sc}$ relating to structural connectivity based gradient of Oldham and Ball (2023) ($r = -0.817$, $p_{SA} < 0.001$) but not transcriptomic patterns ($r = -0.585$, $p_{SA} = 0.227$). Thus, in light of these findings, we conclude that while both of our gradients are reflected in the combined thalamo-cortical structural connectivity and transcriptomic gradient, they show distinct patterns when analyzed separately. Specifically, one gradient aligns with gene expression, and the other is more closely associated with structural connectivity alone. Though somewhat surprising, we believe these differences strengthen the validity of our observed patterns. Both gradients show spatial alignment with the main organizational features identified by an independent research group, while

also demonstrating distinct associations - one with genetic expression and the other with structural connectivity.

Our argument that the thalamus may contain multiple meaningful spatial patterns therefore holds and may extend previous work focussing on a single gradient across modalities. The difference between both patterns may relate to the already discussed differentiation between microstructural patterns (linked to $G1_{sc}$) - and thus in addition related to gene expression, whereas core-matrix patterns (linked to $G2_{sc}$) - relate primarily to structural connectome patterns in HCP.

We have now included this additional observation in our manuscript:

Supplementary Analysis (line 769):

“Association between structural connectivity gradients and thalamic gradients from Oldham and Ball (2023)

To further test the generalizability of our findings, we compared our gradients to those reported by Oldham and Ball (2023), who derived thalamic gradients based on transcriptomic and structural connectivity data. Their source data is available at <https://doi.org/10.1038/s41467-023-41722-8>. $G1_{sc}$ and $G2_{sc}$ from our study were Pearson correlated with Oldham and Ball's main thalamic gradient based on gene expression data joint with structural connectivity (Oldham and Ball, 2023, Fig 2a), and their gradients separately based on gene expression (Oldham and Ball, 2023, Fig S3b), and structural connectivity (Oldham and Ball, 2023, Fig S3c).”

Results (line 198): *“Finally, to test the robustness of our results, we compared the thalamocortical structural connectivity gradients reported here to previous work on joint thalamic gene expression and structural connectivity (Oldham and Ball, 2023). Overall, we found that both $G1_{sc}$ and $G2_{sc}$ were spatially associated with the main gradient reported in Oldham and Ball, (2023; Fig 1) (LH: $G1_{sc}$: $r = 0.478$, $p_{SA} = 0.037$; $G2_{sc}$: $r = -0.701$, $p_{SA} = 0.039$). When examining the relation between $G1_{sc}$ and $G2_{sc}$ and the gradients based on structural connectivity and gene expression separately, we observed a clear differentiation, with our $G1_{sc}$ tending to relate to gene expression (LH: $r = 0.423$, $p_{SA} = 0.054$) but not structural connectivity (LH: $r = -0.061$, $p_{SA} = 0.778$), while $G2_{sc}$ was associated to structural connectivity based gradient of Oldham and Ball (2023) (LH: $r = -0.817$, $p_{SA} < 0.001$) but not transcriptomic patterns (LH: $r = -0.585$, $p_{SA} = 0.227$) (Supplementary Table 3).”*

And we have edited our discussion for a more detailed interpretation of our results.

Discussion (line 373): *“Comparing our work to previously published patterns from Oldham and Ball (2023), we found that both our gradients related to the joint structural connectivity and transcriptomic thalamocortical gradient and differentiate in patterns based on structural connectivity ($G2_{sc}$) and transcriptomic expression ($G1_{sc}$) separately. Though somewhat surprising, we believe these differences strengthen the validity of our observed patterns. Both gradients show spatial alignment with the main organizational features identified by an independent research*

group, while also demonstrating distinct associations - one with genetic expression and the other with structural connectivity. Our argument that the thalamus may contain multiple meaningful spatial patterns therefore holds and may extend previous work focusing on a single gradient across modalities. The difference between both patterns may relate to the already discussed differentiation between microstructural patterns (linked to $G1_{sc}$) - and thus in addition related to gene expression, whereas core-matrix patterns (linked to $G2_{sc}$) - relate primarily to structural connectome patterns in the Human Connectome Project sample.”

	Oldham and Ball (2023)		
	main (gene expression + structural connectivity)	structural connectivity	gene expression
$G1_{sc}$	LH: $r = 0.478$, $p_{SA} = 0.037$	LH: $r = -0.061$, $p_{SA} = 0.778$	LH: $r = 0.423$, $p_{SA} = 0.054$
$G2_{sc}$	LH: $r = -0.701$, $p_{SA} = 0.039$	LH: $r = -0.817$, $p_{SA} < 0.000$	LH: $r = -0.585$, $p_{SA} = 0.227$

Supplementary Table 3: Association between TC structural connectivity gradients and gradients from Oldham and Ball (2023). Pearson correlation between maps corrected for spatial autocorrelation (p_{SA}). Note that the analysis was conducted solely for the left hemisphere, as the data provided by Oldham and Ball (2023) is restricted to this side only. Abbreviations: LH: left hemisphere, sc: structural connectivity

Q 5: Minor point: What are the data points presented in Figure 1F and other 2D plots of $G1$ and $G2$ in other figures? Are they thalamus voxels?

A 5: The data points represent the thalamic voxels in relation to their localization along $G1$ and $G2$. We are happy to clarify and add the missing information to the legend of Fig. 1F, Fig. 2A, B, C, and Supp. 2C.

Figure 1: Thalamocortical Structural Connectivity Gradients. A Normalized group-level structural connectivity matrix resulting from probabilistic tractography computation between thalamic seed-voxels and cortical parcels (i.e., TC). B Affinity matrix derived from group-level structural connectivity matrix using a normalized angle similarity kernel. C Decomposition of affinity matrix into ten gradient components using diffusion map embedding. For each component the corresponding explained variance is displayed. D Gradient loadings of component 1 ($G1_{sc}$) projected on the thalamus (axial planes). The red lines in the glass brain indicate the position of each respective axial slice of the displayed thalamus. E Gradient loadings of component 2 ($G2_{sc}$) projected on the thalamus. Slice positions are congruent to D. F Left: Decoding of $G1_{sc}$ and $G2_{sc}$ based on THOMAS atlas. 2D space framed by $G1_{sc}$ and $G2_{sc}$, color-coded by thalamic subnuclei,

where each datapoint represents a thalamic voxel. Middle and right: Raincloud plots display the gradient loadings of $G1_{sc}$ and $G2_{sc}$ per nucleus and are ordered by median, respectively. All results are presented for the left hemisphere, however, they were similarly replicated in the right thalamus (Supplementary Figure 2). Abbreviations in F: AV: Anterior ventral nucleus, VA: Ventral anterior nucleus, VL_a: Ventral lateral anterior nucleus, VLP: Ventral lateral posterior nucleus, VPL: Ventral posterior lateral nucleus, Pul: Pulvinar nucleus, MGN: Medial geniculate nucleus, CM: Centromedian nucleus, MD: Mediodorsal nucleus, Hb: Habenular nucleus, MTT: Mammillothalamic tract

Figure 2: Contextualization of Gradients with Microstructure and Functional Connectivity. **A** (left) Individual thalamic qT1 values were averaged to create a group-level qT1 map. Note, the inverse relation between qT1 intensity and approximated grey matter myelin. (middle) 2D space framed by $G1_{sc}$ and $G2_{sc}$, color-coded by thalamic group-level qT1 intensity, where data points represent thalamic voxels. (right) Correlation between qT1-intensity and $G1_{sc}$ (upper), and $G2_{sc}$ (lower). **B** (left) Conceptualized representation of the core-matrix framework. Core cells (blue) project in a specific fashion to granular layers of the cerebral cortex, whereas matrix cells (red) innervate superficial cortex layers in a distributed fashion. (middle) 2D space framed by $G1_{sc}$ and $G2_{sc}$, color-coded by the core-matrix difference map with data points representing thalamic voxels. Negative values (blue) of the colormap indicate a higher proportion of core cells, whereas positive values (red) indicate a higher proportion of matrix cells. (right) Correlation between core-matrix difference map and $G1_{sc}$ (upper) and $G2_{sc}$ (lower). **C** (left) Functional connectivity matrix (z-scored) resulting from correlating thalamic voxel and cortical parcel time-series and derived gradient decomposition into 10 components with respective eigenvalues. Principal and secondary components ($G1_{fc}$ and $G2_{fc}$) displayed on the axial thalamus slice (see Supplementary Figure 3 for right hemisphere). (middle) 2D space framed by $G1_{sc}$ and $G2_{sc}$, color-coded by (top) $G1_{fc}$ loadings and (bottom) $G2_{fc}$ loadings. Data points represent thalamic voxels. (right) Correlation between (top) $G1_{fc}$ and structural connectivity gradients $G1_{sc}$ and $G2_{sc}$, and (bottom) $G2_{fc}$ and structural connectivity gradients $G1_{sc}$ and $G2_{sc}$; All results displayed for the left hemisphere.

Supplementary Figure 2: Thalamocortical Structural Connectivity Gradients (RH). **A** Gradient loadings of component 1 ($G1_{sc}$) projected on the thalamus (axial planes). The red lines in the glass brain indicate the position of each respective axial slice of the displayed thalamus. **B** Gradient loadings of component 2 ($G2_{sc}$) projected on the thalamus. Slice positions are congruent to **A**. **C** (left) Decoding of $G1_{sc}$ and $G2_{sc}$ based on THOMAS atlas. 2D space framed by $G1_{sc}$ and $G2_{sc}$, color-coded by thalamic subnuclei, where each data point represents a thalamic voxel. (middle and right) Raincloud plots display the gradient loadings of $G1_{sc}$ and $G2_{sc}$ per nucleus and are ordered by median, respectively. Abbreviations in C: AV: Anterior ventral nucleus, VA: Ventral anterior nucleus, VL_a: Ventral lateral anterior nucleus, VLP: Ventral lateral posterior nucleus, VPL: Ventral posterior lateral nucleus, Pul: Pulvinar nucleus, MGN: Medial geniculate nucleus, CM: Centromedian nucleus, MD: Mediodorsal nucleus, Hb: Habenular nucleus, MTT: Mammillothalamic tract

Reviewer #2 (Remarks to the Author):

The analysis is well-executed, utilizing advanced preprocessing and post-processing techniques to depict the low-dimensional connectivity patterns between the thalamus and cortex, i.e., gradients. However, the results remain speculative due to the partial selection of gradients and the broader association of both gradients with the thalamus nuclei. It is difficult to comprehend the results with respect to the two modes of organization in the depicted gradients (Figure 1) and associated results and discussion.

Q I: Given that the thalamus is significantly affected by a lack of SNR/tSNR, it is essential to assess the quality of the results. Please provide the thalamus SNR values in violin plots for all subjects for the qT1, T1 images, and tSNR maps, as well as of for the DWI and rsfMRI data.

A I: We thank the Reviewer for emphasizing the importance of data quality assessment. In our study, we leverage the publicly available MICA dataset specifically designed for multiscale imaging, which adheres to rigorous quality control standards (see Royer et al., 2022). The data was thoroughly preprocessed; however, we agree that the thalamus, due to its central location deep within the brain, is inherently more susceptible to noise than more superficial regions like the cortex. Our main analysis is based on thalamocortical structural connectivity. Although we begin the seeding within the thalamus, characterized by gray matter (relatively isotropic), we then trace thalamocortical connections in the white matter (relatively anisotropic). Additionally, we used probabilistic tractography which provides more reliable results by sampling from estimates of fiber orientation distributions and therefore its ability to handle uncertainty compared to non-probabilistic methods such as DTI (Behrens et al., 2003).

In response to the Reviewer's request, we present subject-wise thalamic SNR and tSNR for all modalities (T1W, qT1, DWI, and rsfMRI) in a violin plot. All calculations were performed in native space and include data from both hemispheres. For the T1W and qT1 modalities, we calculated the SNR by dividing the signal in each thalamic voxel by the standard deviation of the signal within the thalamus, followed by averaging the voxel-wise SNR values. For the DWI data, we first computed the mean of all preprocessed b0 images and applied the same SNR calculation as described above. For rsfMRI, tSNR was computed by dividing the mean time-series signal in each voxel of the motion-corrected functional data by its standard deviation, followed by averaging the resulting values across all thalamic voxels.

Each datapoint in the violin plot shows the average SNR/tSNR of one subject inside the thalamus. For comparison, we here also report the whole-brain SNR for T1W images and the tSNR for the functional data that were provided with the dataset (calculated using MRIQC, <https://mriqc.readthedocs.io/en/latest/>). The T1W images were only used for coregistration (mean SNR thalamus 14.58, mean SNR total 8.66). The mean SNR of qT1 across subjects was 6.84. We

report the SNR value inside the thalamus based on b0 images (mean SNR thalamus 6.2). More detailed information about imaging quality such as movement parameters in DWI can be gathered from Royer et al., 2022 (see Royer et al., 2022, Figure 3). We further calculated tSNR of functional data (mean tSNR thalamus 38.23, mean tSNR total 47.53).

We acknowledge that the signal-to-noise ratio is lower in the thalamus compared to the cortical surface regions; however, we believe that the comprehensive preprocessing and analytical methods employed in our study enable us to derive meaningful insights from the data.

Furthermore, we also calculated spatial SNR/tSNR maps of the modalities used in our analysis (T1W not shown, since it was only used for coregistration purposes). For DWI, we take the mean of each subject's b0 images and warp this image to MNI space. We report both the mean B0 across all subjects and the spatial SNR maps, calculated by dividing each voxel's mean across subjects by each voxel's standard deviation across subjects. The latter approach is analog used for calculation of qT1 spatial SNR maps. For resting state functional data, we warped the subjects tSNR maps (as computed above) to MNI space and built the mean across subjects resulting in a group-level spatial tSNR map. Following, we evaluated the spatial association between SNR/tSNR and thalamic gradients by correlating (Pearson) the maps with $G1_{sc}$ and $G2_{sc}$ and correcting for spatial autocorrelation. SNR of qT1 (LH: $G1_{sc}$: $r = 0.213$, $p_{SA} = 0.197$; $G2_{sc}$: $r = 0.222$, $p_{SA} = 0.184$) and tSNR of rsfMRI (LH: $G1_{sc}$: $r = 0.334$, $p_{SA} = 0.192$; $G2_{sc}$: $r = 0.158$, $p_{SA} = 0.581$) did not correlate with the gradients. The SNR b0 map correlated with $G1_{sc}$ (LH: $G1_{sc}$: $r = 0.717$, $p_{SA} < 0.000$; $G2_{sc}$: $r = 0.1773$, $p_{SA} = 0.567$). Results are similar in the right hemisphere. Finally, we found that for mean b0 left $G1$ did not significantly correlate (LH: $G1_{sc}$: $r = 0.390$, $p_{SA} = 0.110$; $G2_{sc}$: $r = -0.323$, $p_{SA} = 0.199$), while in the right hemisphere did (RH: $G1_{sc}$: $r = -0.492$, $p_{SA} = 0.008$; $G2_{sc}$: $r = -0.142$, $p_{SA} = 0.624$). Given the lack of consensus on SNR measurement in DWI, particularly for structures like the thalamus in gray matter, we assessed the mean b0 map and b0 SNR map. Typically, SNR in DWI is calculated in white matter tracts rather than in gray matter seed regions. Associations between the first gradient and spatial mean and SNR b0 maps may point to a potential confounder of our gradients and deserve further discussion. B0 refers to the reference image taken

without any diffusion weighting. The thalamic mean b_0 intensity pattern relates to the first gradient but only in the right hemisphere, which does not explain the left hemisphere patterns. By calculating the SNR across subjects the b_0 is here used to measure variability between subjects (std across subjects). This between-subject variability could arise from various factors, including biological differences, scanner distortions, or subject motion. Thus, the relationship between variability of b_0 and the G1 could have both a biological and non-biological interpretation. However, importantly, calculating the gradients is based on the group-level connectivity matrices, underscoring consistency across individuals rather than differences. Moreover, the thalamus serves only as the starting point for tracking connections to the cortex. Tracking itself gave us plausible results in the sense that the nuclei mapped mostly to the expected cortical areas (see Supplementary Fig 7). Overall, we believe that the variation in SNR in b_0 coming from interindividual variability along our axis is an important finding that should be addressed in future studies and might have biological relevance, yet goes beyond our scope. Since our analysis is based on group-level connectivity data, it should not be the cause for our gradients. Moreover, we find structural gradient-related patterns in other modalities (qT1 and functional), where these patterns are clearly unrelated to SNR.

We now add the spatial SNR/tSNR maps to the supplements (Supplementary Figure 9) and discuss it as potential limitations.

Supplementary Analysis (line 779):

“Thalamic SNR and tSNR maps and their association to structural connectivity gradients

Further, we investigated the spatial distribution of (temporal) signal-to-noise ratio (SNR/tSNR) in the thalamus. For DWI, the mean of each subject's b_0 images was computed and warped to MNI space. We report both the mean B_0 across all subjects and the spatial SNR maps, calculated by dividing each voxel's mean across subjects by each voxel's standard deviation across subjects. The latter approach is analog used for the calculation of qT1 spatial SNR maps. For rsfMRI data, tSNR was computed per subject by dividing the mean time-series signal in each voxel of the motion-corrected functional data by its standard deviation, followed by warping the subject's tSNR maps to MNI space and calculating the mean across subjects resulting in a group-level spatial tSNR map. Following, we evaluated the spatial association between SNR/tSNR and thalamic gradients by correlating (Pearson) the maps with $G1_{sc}$ and $G2_{sc}$ and corrected for spatial autocorrelation.”

Supplementary Figure 9: Thalamic spatial SNR/tSNR maps. a) Thalamic mean b0 map averaged across subjects. b) Thalamic b0 SNR map, calculated by dividing each voxel's mean b0 intensity across subjects by its standard deviation across subjects. c) Thalamic qT1 SNR map, calculated by dividing each voxel's mean qT1 across subjects by its standard deviation across subjects. d) Mean rsfMRI tSNR map that was calculated by averaging individual tSNR maps. Individual tSNR maps were computed by dividing the mean time-series signal in each voxel of the motion-corrected functional data by its standard deviation.

	Mean b0 map	SNR B0 map	SNR qT1 map	tSNR rsfMRI map
$G_{1_{sc}}$	LH: $r = 0.390$, $p_{SA} = 0.110$, RH: $r = -0.492$, $p_{SA} = 0.008$	LH: $r = 0.717$, $p_{SA} < 0.000$, RH: $r = 0.724$, $p_{SA} = < 0.000$	LH: $r = 0.213$, $p_{SA} = 0.197$, RH: $r = 0.171$, $p_{SA} = 0.273$	LH: $r = 0.334$, $p_{SA} = 0.192$, RH: $r = 0.245$, $p_{SA} = 0.364$
$G_{2_{sc}}$	LH: $r = -0.323$, $p_{SA} = 0.199$, RH: $r = -0.142$, $p_{SA} = 0.624$	LH: $r = 0.1773$, $p_{SA} = 0.567$, RH: $r = 0.167$, $p_{SA} = 0.595$	LH: $r = 0.222$, $p_{SA} = 0.184$, RH: $r = 0.099$, $p_{SA} = 0.528$	LH: $r = 0.158$, $p_{SA} = 0.581$, RH: $r = 0.160$, $p_{SA} = 0.618$

Supplementary Table 4: Association between TC structural connectivity gradients and thalamic mean b0 map, SNR b0 map, SNR qT1 map and tSNR rsfMRI map. Pearson correlation between maps corrected for spatial autocorrelation (p_{SA}).

Discussion/Limitations (line 518): "It should be noted, the thalamus, being deep in the brain, is a relatively noisy region, and thus signal-to-noise ratio (SNR) must be considered as a potential confounder for our gradient measurements (see Supplementary Figure 9 and Supplementary Table 4). To reduce this impact, we used group-level connectivity matrices, which help to average out individual differences. However, future research may benefit from exploring inter-individual variance, as it could hold biological relevance, though this lies beyond the scope of our current work."

Q II: It is also unclear why only two gradients, explaining 46% of the variance (Figure 1C), were included, without accounting for the rest to illustrate the intricacies between the thalamus and cortex.

A II: We thank the Reviewer for their insightful comment regarding the selection of gradients in our analysis. We focused on the patterns that explained more than 15 % of the variation (46 % in total), which we acknowledge is a subjectively chosen threshold. However, we believe this choice is justified due to the biological interpretability of these gradients, as they represent the main patterns in the relationship between the thalamus and cortex, also present and discussed in other work (Oldham and Ball, 2023; Howell et al., 2024; Yang et al., 2020).

We acknowledge that following the elbow criterion (see Figure 1C), gradients 3 and 4 could also be considered. To address your comment thoroughly, we have added gradients 3 and 4 in the supplementary materials, providing a more comprehensive view of the variance patterns.

Methods (line 667): *“After assessing the amount of explained variance for each gradient, we mapped the resulting first four gradients ($G1_{sc}$ - $G4_{sc}$) onto the thalamic mask. Note that in the following analysis, we focus on the principal ($G1_{sc}$) and secondary gradient ($G2_{sc}$).”*

Results (line 129): *“Additionally, gradients 3 and 4 are presented in the supplementary materials (Supplementary Figure 2).”*

Supplementary Figure 2: Thalamic Cortical Structural Connectivity Gradients. Gradient loadings of components 3 and 4 are projected on the thalamus (axial planes).

For completeness, we added a table in the supplements that shows the correlations of structural connectivity gradients 1 to 10 with the intrathalamic feature maps (i.e., qT1, core-matrix map, functional gradient 1 and 2), further elucidating the relationships between the gradients and the local features and added a section describing the results into the Results and Discussion section of the manuscript. Note that, for the core-matrix map we only focus on the left hemisphere as done in the manuscript due to the small sample size (2 donors) on which the right core-matrix map was grounded.

Results (line 190): "Additionally, to ensure thoroughness, we examined whether structural gradients 3 to 10 were related to the group-level qT1 map, the core-matrix distribution, or TC functional gradients 1 and 2. Overall, we observed mostly spurious effects, except for a bilateral association between $G4_{sc}$ and $G2_{fc}$ (LH: $r = 0.388$, $p_{SA} = 0.013$; RH: $r = 0.367$, $p_{SA} = 0.018$), as well as few unilateral associations such as between $G10_{sc}$ and the core-matrix map (LH: $r = 0.324$, $p_{SA} = 0.020$) (Supplementary Table 1)."

Discussion (line 512): "Though in the current work, we focused on structural gradients 1 and 2, these gradients only explained 46.42 % (LH) of the variance. The first two spatial patterns in structural organization are readably interpretable informed by both our analysis and prior findings (Oldham and Ball, 2023; Howell et al., 2024; Yang et al., 2020; Phillips et al., 2019), and gradients explaining less variance become more difficult to give biological meaning and interpretation. Future research may benefit from examining multiple dimensions concurrently, akin to recent approaches linking cortical modes (Pang et al., 2023)."

	group-level qT1 map	core-matrix difference map	$G1_{fc}$	$G2_{fc}$
$G1_{sc}$	LH: $r = -0.536$, $p_{SA} = 0.038$ RH: $r = -0.594$, $p_{SA} = 0.011$	LH: $r = -0.378$, $p_{SA} = 0.135$	LH: $r = 0.526$, $p_{SA} = 0.044$ RH: $r = 0.564$, $p_{SA} = 0.014$	LH: $r = -0.374$, $p_{SA} = 0.083$ RH: $r = -0.133$, $p_{SA} = 0.532$
$G2_{sc}$	LH: $r = -0.068$, $p_{SA} = 0.873$ RH: $r = 0.119$, $p_{SA} = 0.794$	LH: $r = 0.676$, $p_{SA} = 0.044$	LH: $r = -0.095$, $p_{SA} = 0.872$ RH: $r = -0.177$, $p_{SA} = 0.754$	LH: $r = 0.265$, $p_{SA} = 0.241$ RH: $r = 0.484$, $p_{SA} = 0.016$
$G3_{sc}$	LH: $r = -0.112$, $p_{SA} = 0.411$ RH: $r = -0.122$, $p_{SA} = 0.710$	LH: $r = -0.017$, $p_{SA} = 0.949$	LH: $r = 0.214$, $p_{SA} = 0.062$ RH: $r = 0.290$, $p_{SA} = 0.157$	LH: $r = 0.008$, $p_{SA} = 0.944$ RH: $r = -0.039$, $p_{SA} = 0.806$
$G4_{sc}$	LH: $r = -0.132$, $p_{SA} = 0.461$ RH: $r = -0.078$, $p_{SA} = 0.744$	LH: $r = 0.233$, $p_{SA} = 0.098$	LH: $r = 0.173$, $p_{SA} = 0.305$ RH: $r = 0.079$, $p_{SA} = 0.740$	LH: $r = 0.388$, $p_{SA} = 0.013$ RH: $r = 0.367$, $p_{SA} = 0.018$
$G5_{sc}$	LH: $r = -0.010$, $p_{SA} = 0.967$ RH: $r = 0.043$, $p_{SA} = 0.859$	LH: $r = 0.093$, $p_{SA} = 0.615$	LH: $r = 0.273$, $p_{SA} = 0.136$ RH: $r = 0.223$, $p_{SA} = 0.245$	LH: $r = 0.086$, $p_{SA} = 0.546$ RH: $r = 0.082$, $p_{SA} = 0.592$
$G6_{sc}$	LH: $r = 0.027$, $p_{SA} = 0.917$	LH: $r = -0.193$, $p_{SA} = 0.394$	LH: $r = 0.099$, $p_{SA} = 0.781$	LH: $r = -0.155$, $p_{SA} = 0.358$

	RH: $r = -0.006$, $p_{SA} = 0.976$		RH: $r = 0.028$, $p_{SA} = 0.894$	RH: $r = -0.106$, $p_{SA} = 0.384$
$G7_{sc}$	LH: $r = 0.129$, $p_{SA} = 0.242$ RH: $r = 0.096$, $p_{SA} = 0.414$	LH: $r = -0.068$, $p_{SA} = 0.356$	LH: $r = 0.067$, $p_{SA} = 0.612$ RH: $r = 0.002$, $p_{SA} = 0.987$	LH: $r = -0.002$, $p_{SA} = 0.987$ RH: $r = -0.024$, $p_{SA} = 0.822$
$G8_{sc}$	LH: $r = -0.063$, $p_{SA} = 0.759$ RH: $r = 0.036$, $p_{SA} = 0.828$	LH: $r = 0.081$, $p_{SA} = 0.622$	LH: $r = 0.083$, $p_{SA} = 0.693$ RH: $r = -0.020$, $p_{SA} = 0.927$	LH: $r = 0.248$, $p_{SA} = 0.011$ RH: $r = -0.085$, $p_{SA} = 0.320$
$G9_{sc}$	LH: $r = 0.047$, $p_{SA} = 0.528$ RH: $r = -0.046$, $p_{SA} = 0.678$	LH: $r = -0.004$, $p_{SA} = 0.941$	LH: $r = 0.053$, $p_{SA} = 0.480$ RH: $r = 0.033$, $p_{SA} = 0.789$	LH: $r = 0.159$, $p_{SA} = 0.051$ RH: $r = 0.103$, $p_{SA} = 0.345$
$G10_{sc}$	LH: $r = -0.136$, $p_{SA} = 0.417$ RH: $r = 0.081$, $p_{SA} = 0.696$	LH: $r = 0.324$, $p_{SA} = 0.020$	LH: $r = 0.053$, $p_{SA} = 0.480$ RH: $r = -0.095$, $p_{SA} = 0.662$	LH: $r = -0.028$, $p_{SA} = 0.830$ RH: $r = -0.175$, $p_{SA} = 0.251$

Supplementary Table 1: Association between TC structural connectivity gradients 1 to 10 and thalamic maps (qT1 group-level maps, core-matrix maps, TC functional connectivity gradient 1 and 2). Pearson correlation between maps corrected for spatial autocorrelation (p_{SA}). Note that for the core-matrix map, only results of the left hemisphere are reported due to the small sample size on which the right core-matrix map is based. Abbreviations: LH: left hemisphere, RH: right hemisphere, sc: structural connectivity, fc: functional connectivity.

Q III: Figure 1F, which decodes based on the THOMAS atlas, shows a diffuse and sparse association of nuclei in both gradients, resulting in a non-specific nuclei association, contrary to the discussed axis organization. It is advisable to discuss the findings in a broader context while critically examining the observations. In the current stage it is difficult to comprehend the axis organization and its biological meaning as well the differences in the structural, functional and structural covariance.

A III: We thank the Reviewer for this comment that helps us to clarify our discussion. We will first address the reference to Figure 1F and secondly discuss the findings in a broader context:

In Figure 1F, we utilized a widely recognized atlas of thalamic nuclei to integrate established organizational frameworks of the thalamus with our gradients for decoding. To visualize the spatial distribution of nuclei within the gradient dimensions, we plotted the locations of thalamic voxels in a 2D space defined by gradients 1 and 2 color-coding the voxels according to their affiliated nuclei. The density of the data points in the scatter plot is influenced by the threshold applied to the input connectivity matrix, where higher thresholds (e.g., 0.9) result in less diffuse patterns. We applied a threshold of 0.75, which is well-suited for structural connectivity data. We observe that most nuclei tend to localize at specific positions along either both gradients (e.g., VPL, AV) or at least one gradient (e.g., Pulvinar at G2). To decompose whether there is a meaningful ordering of nuclei, we then ordered them based on the median of gradient loading for G1 and G2 separately. We acknowledge that certain nuclei, such as the Pulvinar, are dispersed along G1, limiting our ability to confidently describe a clear pattern of nuclei organization along this axis. However in G1, we see a trend of nuclei that project to association cortices and are considered as higher order nuclei (AV, MTT (connection to mammillary bodies), Hb, MD, VA, Pul) at one pole and nuclei that are involved in sensorimotor processes at the other pole of the gradient (with VLP, VPL, VL_a projecting to somatomotor regions and MGN projecting to auditory cortex). CM is a very small nucleus which is intralaminar and classified as higher order nucleus but also projects to motor regions (Remore et al., 2023; Ilyas et al., 2019). This pattern is also reflected at the cortex level (see Figure 3A), where the projection of G1 onto the cerebral cortex distinguishes between limbic regions and somatomotor regions. For G2 the decoding with the nuclei did not result in a clear pattern.

Overall, we found two axes of organization that are in coherence with underlying microstructural patterns (myelin, core-matrix), functional connectivity, and to some extent structural covariance which is a proxy for shared genetic patterning and maturation (Alexander-Bloch et al., 2013). Studies in mice have shown that the transcriptional profiling of thalamic neurons resulted in a medial-to-lateral pattern, with a topological shift overspanning nuclei borders. This axis reflects the gene expression pattern of voltage-gated ion channels, neurotransmitter modulator receptors, but also varying electrophysiological properties, such as action potential width (Phillips et al., 2019). To that end, our study accesses human thalamic organization at the macroscale with the

interesting finding of a medial-to-lateral- gradient and combining several modalities. Moreover, there are numerous studies showing gradients in the cortex in different modalities, linked to genetic, maturational, structural and functional patterning (Bernhardt et al., 2022; Huntenburg et al., 2018; Valk et al., 2020; Vos de Wael et al., 2021; Paquola et al., 2019). Given the strong connectivity between the thalamus and the cortex, it is likely that the gradient patterns of the cortex may also be present in the thalamus - though causation is unclear. Notably, each of the modalities have a slightly different biological origin: structural connectivity relates to white matter tracts, functional connectivity to more indirect correlations of bold time-series, and quantitative T1 and core/matrix more reflect more local patterning. While we and others have shown these features are to some extent interrelated, we also put forward the hypothesis that different features show differentiation into multiple biological organizational dimensions. This differentiation into multiple dimensions may be reconcilable with the existence of different nuclei, and the differing definition of their boundaries depending on the modality investigated.

We now elaborate on these insights in the revised manuscript, emphasizing their significance and how they contribute to our understanding of thalamic organization:

Discussion (line 303): *“Overall, we identified two axes of organization that align with microstructural patterns (myelin, core-matrix), functional connectivity, and, to some extent, structural covariance, which serves as a proxy for shared genetic patterning and maturation (Alexander-Bloch et al., 2013). In particular, our findings revealed a principal medial-to-lateral thalamic axis/gradient based on thalamocortical white matter connectivity. To link the gradients to their underlying thalamic nuclei composition, we used a well-established atlas for decoding (Saranathan et al., 2021) and observed that most nuclei tend to localize at specific positions along either both gradients (e.g., VPL, AV) or at least one (e.g., Pulvinar along $G2_{sc}$). While nuclei like the Pulvinar showed dispersion along the principal axis, limiting clear pattern identification along this axis, a trend emerged in $G1_{sc}$, where nuclei projecting to association cortices (higher-order nuclei such as AV, MD, VA) cluster at the medial pole of the gradient, and those projecting to sensorimotor cortex (e.g., VPL, VLa, MGN, CM) appeared at the lateral pole. Note, the CM, though a small intralaminar nucleus, is classified as a higher-order nucleus but also projects to motor regions (Remore et al., 2023; Ilyas et al., 2019). We derived that main variations in TC connectivity profiles map an axis differentiating sensorimotor-projecting regions and transmodal regions, which is also consistent with the pattern observed when the principal axis was projected onto the cortex, revealing a clear division between limbic and somatomotor cortical regions. Our observations broadly aligned with recent work using a joint analysis of TC structural connectivity and gene expression data (Oldham and Ball, 2023) that reported a similar medial-to-lateral axis, with the difference that in our case the medial apex tended to be more pronounced to the center of the thalamus. The hierarchical representation of projection patterns was also found in the mouse thalamus after computing an organizational axis based on tract tracing and gene expression data, spanning from somatosensory regions to lateral/frontal regions and may suggest the hypothesis that this axis is phylogenetically-conserved (Oldham and Ball, 2023). Further, the medial-to-lateral axis may originate from the spatiotemporal development of the brain. It has been shown that the formation of connectivity-based subdivisions of the human thalamus expanded from lateral*

to medial portions during the perinatal period (Zheng et al., 2023). Moreover, a study in mice has shown a thalamic medial-to-lateral pattern, with a topological shift across nuclei borders in gene expression and electrophysiological properties (Phillips et al., 2019). Next to a medial-to-lateral axis, we observed a second axis of organization with one apex located at the medial-anterior and posterior pole of the thalamus, and the opposite apex intersecting the thalamus from anterior-lateral to central-medial. Decoding of the secondary gradient with the atlas did not yield a clear pattern and segregated MGN, VPL, AV, Pulvinar, CM, and VLP from Hb, MD, MTT, VA, VLa regions on top of the differentiation observed in the first gradient. [...]"

Discussion (line 369): "[...] While demonstrating that there is an overlap of organizational principles across modalities, differences between patterns of structural and functional connectivity are expected due to method-specificities (i.e., functional connectivity arising not only from direct but also indirect connections). Each modality in our study has a distinct methodological approach that may hint at a differentiable biological origin: structural connectivity reflects white matter tracts, functional connectivity relates to correlations between bold time-series, thought to reflect direct and indirect connectivity, and quantitative T1 reflects local patterning of microstructure, linked to myelination (Weißkopf et al., 2021). While these features are ultimately interrelated to some extent, we propose that they may reflect differentiation into distinct biological pathways, which could align with the varied definitions of nuclei boundaries across different modalities.[...]"

Discussion (line 404): "[...] Our study accessed human thalamic organization at the macroscale, revealing two gradients and combining several modalities. Numerous studies have also demonstrated cortical gradients across genetic, maturational, structural, and functional dimensions (Bernhardt et al., 2022; Huntenburg et al., 2018; Valk et al., 2020; Vos de Wael et al., 2021; Paquola et al., 2019), suggesting that similar gradient patterns in the thalamus may reflect cortical connectivity, though causation remains unclear."

Q IV: The thalamus comprises smaller nuclei, and fully understanding their representation in a gradient ideally requires a much higher resolution fMRI scan than the 3mm isotropic resolution used in this analysis.

A IV: We thank the Reviewer and agree that this is an important note. The spatial resolution may limit the precise representation of thalamic substructures within the gradient. Some thalamic nuclei are indeed very small; for instance, histological studies (e.g., Morel) show that the thalamus can be subdivided into more than 30 nuclei, many of which cannot be fully captured with the resolution of (our) MRI data (e.g., DWI at 1.6 mm, qT1 at 0.8 mm, and MRI at 3 mm). For our analysis, we use the MNI standard space thomas atlas which is created for MRI data and comes in 0.5 mm resolution. To decode the gradient, we downsample it to the same resolution as our structural connectivity results (2 mm resolution), which comes with the disadvantage of losing spatial precision. Therefore, especially small nuclei such as the Hb should be interpreted with caution in this context. We will add this to our limitation discussion/limitations section. We also note that

future studies utilizing ultra-high-resolution scans could provide more granular insights into the thalamic nuclei, enhancing our understanding beyond what is possible with the current dataset.

Discussion (line 496): *“To decode the gradients, we use the THOMAS atlas (Su et al., 2019; Saranathan et al., 2021) to subdivide the thalamus into its constituent nuclei. However, it is important to note that the resolution of our data imposes limitations on the spatial precision of this subdivision, particularly for smaller nuclei, such as the habenular (Hb) and medial geniculate nucleus (MGN). Therefore, conclusions based on the atlas should be interpreted with caution. Utilizing 7T data in future studies is necessary to provide more granular insights of thalamic organization (Cabalo et al., 2024)”*

Q V: If the THOMAS thalamus segmentation method was applied to individual subjects, the nuclei volumes would reveal inter-individual differences. Therefore, it is important to analyze these differences in addition to the group-level analysis.

A V: We thank the Reviewer for this insightful and constructive comment. In the current study, we have focused primarily on group-level analysis by registering all native space images to MNI standard space and then generating group-level maps. We acknowledge that analyzing inter-individual differences of thalamic organization is indeed highly relevant for future studies, however, such an exploration falls outside the scope of the present work. Instead, we focused on mapping generalizable features of thalamic organization, aiming to uncover generalizable patterns that may serve as a foundation for future studies examining more fine-grained individual variability. Our findings suggest that the gradients of interest are relatively consistent across individuals (see Supplementary Figure 5), reinforcing the validity of group-level conclusions. The THOMAS atlas was specifically utilized for topological decoding rather than volumetric assessment. Our intention was to investigate how the structural gradients correspond to the spatial arrangement of thalamic nuclei, with the primary goal being to investigate the relationship between these gradients and nuclei locations, rather than to focus on individual-level volumetric differences.

In conclusion, while the current study focuses on group-level patterns, we consider inter-individual differences to be an important angle for future studies, which could deepen our understanding of how these gradients are expressed on an individual basis.

The following paragraph was added to the Discussion:

Discussion (line 503): *“Additionally, the present study focuses on group-level analysis, aiming to reveal generalizable organizational principles of the thalamus. While this approach provides valuable insights into group-level patterns, individual-level variability in thalamic structure remains unexplored in this work. Future research could address this limitation by incorporating subject-specific analyses to uncover how these gradients manifest at an individual level.”*

References

- Alexander-Bloch, Raznahan, A., Bullmore, E., Giedd, J., 2013. The Convergence of Maturation Change and Structural Covariance in Human Cortical Networks. *Journal of Neuroscience* 33, 2889–2899. <https://doi.org/10.1523/JNEUROSCI.3554-12.2013>
- Behrens, T.E.J., Johansen-Berg, H., Woolrich, M.W., Smith, S.M., Wheeler-Kingshott, C.A.M., Boulby, P.A., Barker, G.J., Sillery, E.L., Sheehan, K., Ciccarelli, O., Thompson, A.J., Brady, J.M., Matthews, P.M., 2003. Non-invasive mapping of connections between human thalamus and cortex using diffusion imaging. *Nat Neurosci* 6, 750–757. <https://doi.org/10.1038/nn1075>
- Bernhardt, B.C., Smallwood, J., Keilholz, S., Margulies, D.S., 2022. Gradients in brain organization. *NeuroImage* 251, 118987. <https://doi.org/10.1016/j.neuroimage.2022.118987>
- Cabalo, D.G., Leppert, I.R., Thevakumaran, R., DeKraker, J., Hwang, Y., Royer, J., Kebets, V., Tavakol, S., Wang, Y., Zhou, Y., Benkarim, O., Eichert, N., Paquola, C., Tardif, C.L., Rudko, D., Smallwood, J., Rodriguez-Cruces, R., Bernhardt, B.C., 2024. Multimodal precision neuroimaging of the individual human brain at ultra-high fields. <https://doi.org/10.1101/2024.06.17.596303>
- Halassa, M.M., Sherman, S.M., 2019. Thalamocortical Circuit Motifs: A General Framework. *Neuron* 103, 762–770. <https://doi.org/10.1016/j.neuron.2019.06.005>
- Howell, A.M., Warrington, S., Fonteneau, C., Cho, Y.T., Sotiropoulos, S.N., Murray, J.D., Anticevic, A., 2024. The spatial extent of anatomical connections within the thalamus varies across the cortical hierarchy in humans and macaques. *eLife* 13. <https://doi.org/10.7554/eLife.95018.1>
- Huntenburg, J.M., Bazin, P.-L., Margulies, D.S., 2018. Large-Scale Gradients in Human Cortical Organization. *Trends in Cognitive Sciences* 22, 21–31. <https://doi.org/10.1016/j.tics.2017.11.002>
- Iglesias, J.E., Insausti, R., Lerma-Usabiaga, G., Bocchetta, M., Van Leemput, K., Greve, D.N., van der Kouwe, A., Fischl, B., Caballero-Gaudes, C., Paz-Alonso, P.M., 2018. A probabilistic atlas of the human thalamic nuclei combining ex vivo MRI and histology. *NeuroImage* 183, 314–326. <https://doi.org/10.1016/j.neuroimage.2018.08.012>
- Ilyas, A., Pizarro, D., Romeo, A.K., Riley, K.O., Pati, S., 2019. The centromedian nucleus: Anatomy, physiology, and clinical implications. *Journal of Clinical Neuroscience* 63, 1–7. <https://doi.org/10.1016/j.jocn.2019.01.050>
- Jones, E.G., 1998. Viewpoint: the core and matrix of thalamic organization. *Neuroscience* 85, 331–345. [https://doi.org/10.1016/S0306-4522\(97\)00581-2](https://doi.org/10.1016/S0306-4522(97)00581-2)
- Kim, C.N., Shin, D., Wang, A., Nowakowski, T.J., 2023. Spatiotemporal molecular dynamics of the developing human thalamus. *Science* 382, eadf9941. <https://doi.org/10.1126/science.adf9941>
- Margulies, D.S., Ghosh, S.S., Goulas, A., Falkiewicz, M., Huntenburg, J.M., Langs, G., Bezgin, G., Eickhoff, S.B., Castellanos, F.X., Petrides, M., Jefferies, E., Smallwood, J., 2016. Situating the default-mode network along a principal gradient of macroscale cortical organization. *Proc. Natl. Acad. Sci. U.S.A.* 113, 12574–12579. <https://doi.org/10.1073/pnas.1608282113>
- Morel, A., Magnin, M., Jeanmonod, D., 1997. Multiarchitectonic and stereotactic atlas of the human thalamus. *J. Comp. Neurol.* 387, 588–630. [https://doi.org/10.1002/\(SICI\)1096-9861\(19971103\)387:4<588::AID-CNE8>3.0.CO;2-Z](https://doi.org/10.1002/(SICI)1096-9861(19971103)387:4<588::AID-CNE8>3.0.CO;2-Z)
- Najdenovska, E., Alemán-Gómez, Y., Battistella, G., Descoteaux, M., Hagmann, P., Jacquemont, S., Maeder, P., Thiran, J.-P., Fornari, E., Bach Cuadra, M., 2018. In-vivo probabilistic atlas of human thalamic nuclei based on diffusion-weighted magnetic

- resonance imaging. *Sci Data* 5, 180270. <https://doi.org/10.1038/sdata.2018.270>
- Oldham, S., Ball, G., 2023. A phylogenetically-conserved axis of thalamocortical connectivity in the human brain. *Nat Commun* 14, 6032. <https://doi.org/10.1038/s41467-023-41722-8>
- Oldham, S., Ball, G., 2022. A phylogenetically-conserved axis of thalamocortical connectivity in the human brain (preprint). *Neuroscience*. <https://doi.org/10.1101/2022.11.15.516574>
- Pang, J.C., Aquino, K.M., Oldehinkel, M., Robinson, P.A., Fulcher, B.D., Breakspear, M., Fornito, A., 2023. Geometric constraints on human brain function. *Nature* 618, 566–574. <https://doi.org/10.1038/s41586-023-06098-1>
- Paquola, C., Hong, S.-J., 2023. The Potential of Myelin-Sensitive Imaging: Redefining Spatiotemporal Patterns of Myeloarchitecture. *Biological Psychiatry, Making Connections: Biological Mechanisms of Human Brain (Dys)connectivity* 93, 442–454. <https://doi.org/10.1016/j.biopsych.2022.08.031>
- Paquola, C., Vos De Wael, R., Wagstyl, K., Bethlehem, R.A.I., Hong, S.-J., Seidlitz, J., Bullmore, E.T., Evans, A.C., Misic, B., Margulies, D.S., Smallwood, J., Bernhardt, B.C., 2019. Microstructural and functional gradients are increasingly dissociated in transmodal cortices. *PLoS Biol* 17, e3000284. <https://doi.org/10.1371/journal.pbio.3000284>
- Park, S., Haak, K.V., Oldham, S., Cho, H., Byeon, K., Park, B., Thomson, P., Chen, H., Gao, W., Xu, T., Valk, S., Milham, M.P., Bernhardt, B., Di Martino, A., Hong, S.-J., 2024. A shifting role of thalamocortical connectivity in the emergence of cortical functional organization. *Nat Neurosci* 1–11. <https://doi.org/10.1038/s41593-024-01679-3>
- Phillips, J.W., Schulmann, A., Hara, E., Winnubst, J., Liu, C., Valakh, V., Wang, L., Shields, B.C., Korff, W., Chandrashekar, J., Lemire, A.L., Mensh, B., Dudman, J.T., Nelson, S.B., Hantman, A.W., 2019. A repeated molecular architecture across thalamic pathways. *Nat Neurosci* 22, 1925–1935. <https://doi.org/10.1038/s41593-019-0483-3>
- Remore, L.G., Rifi, Z., Nariyai, H., Eliashiv, D.S., Fallah, A., Edmonds, B.D., Matsumoto, J.H., Salamon, N., Tolossa, M., Wei, W., Locatelli, M., Tsolaki, E.C., Bari, A.A., 2023. Structural connections of the centromedian nucleus of thalamus and their relevance for neuromodulation in generalized drug-resistant epilepsy: insight from a tractography study. *Ther Adv Neurol Disord* 16, 17562864231202064. <https://doi.org/10.1177/17562864231202064>
- Roy, D.S., Zhang, Y., Halassa, M.M., Feng, G., 2022. Thalamic subnetworks as units of function. *Nat Neurosci* 25, 140–153. <https://doi.org/10.1038/s41593-021-00996-1>
- Royer, J., Rodríguez-Cruces, R., Tavakol, S., Larivière, S., Herholz, P., Li, Q., Vos de Wael, R., Paquola, C., Benkarim, O., Park, B., Lowe, A.J., Margulies, D., Smallwood, J., Bernasconi, A., Bernasconi, N., Frauscher, B., Bernhardt, B.C., 2022. An Open MRI Dataset For Multiscale Neuroscience. *Sci Data* 9, 569. <https://doi.org/10.1038/s41597-022-01682-y>
- Sandrone, S., Aiello, M., Cavaliere, C., Thiebaut de Schotten, M., Reimann, K., Troakes, C., Bodi, I., Lacerda, L., Monti, S., Murphy, D., Geyer, S., Catani, M., Dell'Acqua, F., 2023. Mapping myelin in white matter with T1-weighted/T2-weighted maps: discrepancy with histology and other myelin MRI measures. *Brain Struct Funct* 228, 525–535. <https://doi.org/10.1007/s00429-022-02600-z>
- Saranathan, M., Iglehart, C., Monti, M., Tourdias, T., Rutt, B., 2021. In vivo high-resolution structural MRI-based atlas of human thalamic nuclei. *Sci Data* 8, 275. <https://doi.org/10.1038/s41597-021-01062-y>
- Sherman, S.M., 2012. Thalamocortical interactions. *Current Opinion in Neurobiology, Microcircuits* 22, 575–579. <https://doi.org/10.1016/j.conb.2012.03.005>
- Smallwood, J., Bernhardt, B.C., Leech, R., Bzdok, D., Jefferies, E., Margulies, D.S., 2021. The default mode network in cognition: a topographical perspective. *Nat Rev Neurosci* 22, 503–513. <https://doi.org/10.1038/s41583-021-00474-4>
- Su, J.H., Thomas, F.T., Kasoff, W.S., Tourdias, T., Choi, E.Y., Rutt, B.K., Saranathan, M., 2019.

- Thalamus Optimized Multi Atlas Segmentation (THOMAS): fast, fully automated segmentation of thalamic nuclei from structural MRI. *NeuroImage* 194, 272–282. <https://doi.org/10.1016/j.neuroimage.2019.03.021>
- Valk, S.L., Xu, T., Margulies, D.S., Masouleh, S.K., Paquola, C., Goulas, A., Kochunov, P., Smallwood, J., Yeo, B.T.T., Bernhardt, B.C., Eickhoff, S.B., 2020a. Shaping brain structure: Genetic and phylogenetic axes of macroscale organization of cortical thickness. *Sci. Adv.* 6, eabb3417. <https://doi.org/10.1126/sciadv.abb3417>
- Valk, S.L., Xu, T., Margulies, D.S., Masouleh, S.K., Paquola, C., Goulas, A., Kochunov, P., Smallwood, J., Yeo, B.T.T., Bernhardt, B.C., Eickhoff, S.B., 2020b. Shaping brain structure: Genetic and phylogenetic axes of macroscale organization of cortical thickness. *Science Advances* 6, eabb3417. <https://doi.org/10.1126/sciadv.abb3417>
- Vos de Wael, R., Royer, J., Tavakol, S., Wang, Y., Paquola, C., Benkarim, O., Eichert, N., Larivière, S., Xu, T., Misic, B., Smallwood, J., Valk, S.L., Bernhardt, B.C., 2021. Structural Connectivity Gradients of the Temporal Lobe Serve as Multiscale Axes of Brain Organization and Cortical Evolution. *Cereb Cortex* 31, 5151–5164. <https://doi.org/10.1093/cercor/bhab149>
- Weiskopf, N., Edwards, L.J., Helms, G., Mohammadi, S., Kirilina, E., 2021. Quantitative magnetic resonance imaging of brain anatomy and in vivo histology. *Nat Rev Phys* 3, 570–588. <https://doi.org/10.1038/s42254-021-00326-1>
- Yang, S., Meng, Y., Li, J., Li, B., Fan, Y.-S., Chen, H., Liao, W., 2020. The thalamic functional gradient and its relationship to structural basis and cognitive relevance. *NeuroImage* 218, 116960. <https://doi.org/10.1016/j.neuroimage.2020.116960>
- Zheng, W., Zhao, L., Zhao, Z., Liu, T., Hu, B., Wu, D., 2023. Spatiotemporal Developmental Gradient of Thalamic Morphology, Microstructure, and Connectivity from the Third Trimester to Early Infancy. *J. Neurosci.* 43, 559–570. <https://doi.org/10.1523/JNEUROSCI.0874-22.2022>